# Scalable 3D Captioning with Pretrained Models

**Tiange Luo**[1,*]   **Chris Rockwell**[1,*]   **Honglak Lee**[1,2,†]   **Justin Johnson**[1,†]
[1]University of Michigan    [2]LG AI Research

## Abstract

We introduce Cap3D, an automatic approach for generating descriptive text for 3D objects. This approach utilizes pretrained models from image captioning, image-text alignment, and LLM to consolidate captions from multiple views of a 3D asset, completely side-stepping the time-consuming and costly process of manual annotation. We apply Cap3D to the recently introduced large-scale 3D dataset, Objaverse, resulting in 785k 3D-text pairs. Our evaluation, conducted using 41k human annotations from the same dataset, demonstrates that Cap3D surpasses human-authored descriptions in terms of quality, cost, and speed. Through effective prompt engineering, Cap3D rivals human performance in generating geometric descriptions on 17k collected annotations from the ABO dataset. Finally, we finetune text-to-3D models on Cap3D and human captions, and show Cap3D outperforms; and benchmark the SOTA including Point·E, Shap·E, and DreamFusion. Our data, code, and finetuned models can be found at https://cap3d-um.github.io/.

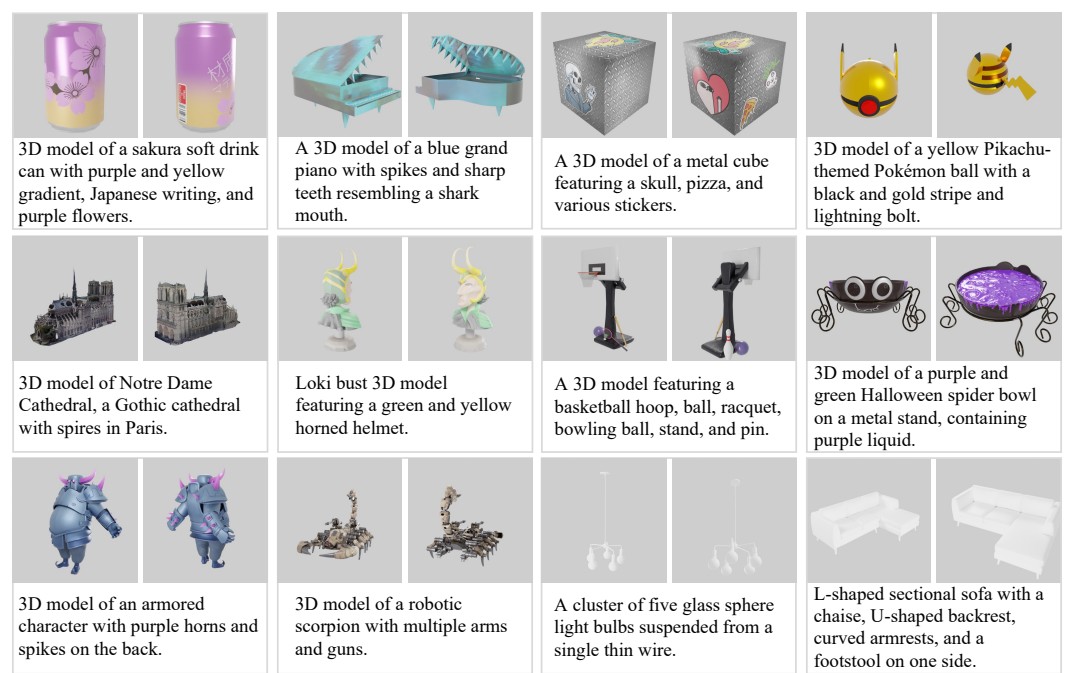

| | | | |
|---|---|---|---|
| 3D model of a sakura soft drink can with purple and yellow gradient, Japanese writing, and purple flowers. | A 3D model of a blue grand piano with spikes and sharp teeth resembling a shark mouth. | A 3D model of a metal cube featuring a skull, pizza, and various stickers. | 3D model of a yellow Pikachu-themed Pokémon ball with a black and gold stripe and lightning bolt. |
| 3D model of Notre Dame Cathedral, a Gothic cathedral with spires in Paris. | Loki bust 3D model featuring a green and yellow horned helmet. | A 3D model featuring a basketball hoop, ball, racquet, bowling ball, stand, and pin. | 3D model of a purple and green Halloween spider bowl on a metal stand, containing purple liquid. |
| 3D model of an armored character with purple horns and spikes on the back. | 3D model of a robotic scorpion with multiple arms and guns. | A cluster of five glass sphere light bulbs suspended from a single thin wire. | L-shaped sectional sofa with a chaise, U-shaped backrest, curved armrests, and a footstool on one side. |

Figure 1: Cap3D provides detailed descriptions of 3D objects by leveraging pretrained models in captioning, alignment, and LLM to consolidate multi-view information. Two views of 3D objects are shown here, Cap3D uses eight. Additional examples are available in Appendix B.

---

* joint first authorship; † equal advising

37th Conference on Neural Information Processing Systems (NeurIPS 2023) Track on Datasets and Benchmarks.

Table 1: Cap3D is **better, cheaper, and faster** than crowdsourced annotation. Use 36k responses across 22k objects for A/B testing; 8A40s on a cloud platform for speed and cost computations.

| Method | A/B Human Testing Win % (Tie %) | Cost per 1k Objects | Annotation Speed | 1k Objects Cost Breakdown | |
|--------|------------------------------|---------------------|------------------|---------------------------|---|
| | | | | BLIP2 | $3.79 |
| | | | | CLIP | $0.38 |
| | | | | GPT4 | $4.18 |
| Human | 37.8% ± 0.5% (9.5%) | $87.18 | 1.4k / day | | |
| Cap3D | **52.3% ± 0.5% (9.5%)** | **$8.35** | **65k / day** | Cap3D Total Cost | **$8.35** |

# 1 Introduction

Text-conditioned 3D synthesis [1–3] could revolutionize the creation process of 3D assets, impacting various sectors, including 3D design, virtual reality [4], film [5], robotics [6, 7], and autonomous driving [8]. However, challenges persist, namely the high cost of 3D asset creation and the scarcity of high-quality captions for 3D assets. Objaverse [9] takes a step towards this as the first public large-scale 3D object dataset. Unfortunately, while objects contain paired metadata, these do not serve as informative captions, as shown in Table 3. In contrast with 3D, a plethora of high-quality text-image paired data is publicly available [10–14]. This data has led to incredible recent progress in image-text learning [15–18], text-conditioned image synthesis [19–24], and image captioning [25–29].

In this work, we present Cap3D, a method to automate 3D object annotation. Our key insight is to leverage the abundance of knowledge in pretrained image-text models to remedy the lack of existing 3D-text data. The core of our data collection process is to apply an image captioning model (BLIP2 [29]) to a set of 3D asset renders, use an image-text alignment model (CLIP [16]) to filter captions, and apply a language model (GPT4 [30]) to fuse the filtered captions across views. Critically, the models we apply are pretrained on varied and large-scale text-image [11–13, 31–33], and text [34], data; and approach complementary problems. As a result, each model adds additional value to the framework, as we show in Table 3.

Cap3D is agnostic to 3D asset sources and can be effectively scaled to larger extents with increased 3D assets and computational resources. In this paper, we apply it primarily to Objaverse, gathering a dataset of 785k 3D-text pairs. Through object rendering and captioning, we enable ethical filtering of 3D objects via both image and text, as detailed in § 3.2. We publicly release all of our collected data including automated and human-annotated captions, along with associated Point Clouds and Rendered Images, at huggingface.co/datasets/tiange/Cap3D. The dataset is released under ODC-By 1.0 license. We also released trained models and code for replicating the benchmark table.

We validate our collection approach by collecting over 50k crowdsourced captions on over 40k objects. We conduct human evaluations and show on Objaverse that our automated captions are superior to crowdsourced captions in quality, cost, and speed (Table 1, details in Appendix A). Specifically, it is preferred 35% more often by humans, costs more than 10 times less, and is over 40 times faster, assuming only 8A40 GPUs. We also test the limits of automated captioning. We consider a separate task of captioning geometry (as shown in Figure 1 bottom-right) using ABO, a dataset of 3D models with complex geometries [35]. Shown in Table 4, our automated captioning underperforms humans. However, by formulating description as a question answering task (detailed in § 3.1), we show stronger performance compared to crowdsourced workers. This result shows the ability of our method to adapt beyond traditional captioning and still be highly competitive.

Finally, our high-quality gathered 3D-text dataset enables us to train and validate large-scale text-to-3D models. In §5.3, we evaluate several state-of-the-art methods on Objaverse out-of-the box, including Point·E, Shap·E, DreamFields, and DreamFusion. Finetuning on our data typically shows meaningful improvements, demonstrating the value of the collected dataset. In addition, we show our automatically collected captions yield better finetuning performance than human captions – even at the same scale. At full scale, finetuning is further boosted.

# 2 Related Work

Obtaining 3D-text pairs at scale is challenging, and we take inspiration from image-text datasets and methods when approaching this task.

**Image-Text Data and Modeling.** Early image captioning [36–38] and text-image representation learning methods [39–41] were built using CNNs [42–44] and LSTMs [45, 46], leveraging human-annotated datasets [31–33, 47]. Text-to-image methods used similar datasets, and relied on GANs [48, 49] and VQVAEs [19, 50–52]. The advent of semi-automated image-text collection has enabled successful scaling of datasets [10–14] and models [25–28]. Transformer-based architectures [16, 53, 54] and diffusion models [55–60] have scaled best to large data; we employ transformer-based methods through our captioning process and adopt diffusion models for text-to-3D experiments.

Training models upon large datasets and using the corresponding trained models to filter larger data has led to datasets of rapidly increasing size [13, 14]. In addition to filtering, trained models have been used to annotate new data with high-quality [61]. We take this approach, captioning rendered views with BLIP2 [29], refining with CLIP [16, 62], and summarizing with GPT4 [63]; all of which are trained on large datasets, including [11–13, 31–33]. Concurrent works [64–66] use automated captioning on 2D images using an older system [67] or based upon metadata [66, 68].

**3D-Text Data and Modeling.** Until recently, 3D data was of relatively small scale ($\sim 50$k objects) [69–72]. Labeled 3D-text data was scarce, relying on human annotation, and typically limited to ShapeNet [69] chairs [73] or tables and chairs [74, 75], and ScanNet [76, 77]. This enabled prior work to undertake the task of 3D captioning [78–80] or text-to-3D [74, 78, 81–84] at small scale. Methods that approached text-to-3D would sometimes avoid 3D supervision entirely [3, 85–87], leading to slow generation due to many optimization steps. We annotate a small-scale dataset containing 3D furniture, ABO [35], to evaluate the ability of Cap3D to specify fine-grained geometry.

Objaverse [9] introduced a diverse set of objects over 10 times the size of the prior largest public 3D dataset [69]. This data is our primary captioning focus, and we associate a single caption with each object in Objaverse after filtering. Concurrent works [66, 88] gather text associated with Objaverse, but do not fuse captions across views [88] or rely upon metadata [66], and do not approach text-to-3D.

The concurrent studies 3DGen [2] learns text and image to 3D on Objaverse; Point·E [89] and Shap·E [90] learn text-to-3D models on a large-scale 3D dataset, but none have fully disclosed their code or data. Point·E involves two variants and released a text-to-3D model and a text-to-image-to-3D model by finetuning GLIDE [23] and training an image-to-point cloud diffusion model [91]. Other recent works [92, 93] also focus on scaled image-3D generation. We show finetuning on our captions improves Point·E performance despite having already been trained on large amounts of Internet data.

## 3 Method

### 3.1 Captioning Process

Our task is to produce a single descriptive caption given a 3D asset. Our proposed method, Cap3D, employs a four-step process. First, we render a set of 2D views for each 3D object. Next, we apply image captioning to achieve preliminary descriptions. As these captions may contain inaccuracies, an image-text alignment model, CLIP, is introduced in the third step to rectify errors. Finally, an LLM is employed to unify captions from various perspectives, creating a comprehensive caption. This process is shown in Figure 2 and detailed below.

**Object Rendering**: We render using Blender at $512{\times}512$ from $M = 8$ high-information camera angles rotating horizontally around the object, with two slightly below and the rest slightly above the object, to cover all the object details. The reason we prefer multiple views is a forward-facing view may miss self-occluded object details (e.g. Figure 1 row 1) or face strange appearance and/or lighting. In contrast, multiple views will see much of the object from different viewpoints, increasing the number of chances for a captioning model to predict objects in detail. For instance, in Figure 2, the back view 1 identifies the "yellow handle", which is barely visible in forward view $M$.

**Image Captioning:** We use BLIP2 [29] for captioning, selecting the largest pretrained model adapting ViT-G [54, 94] image encoder and FlanT5XXL [95] text encoder. We generate $N = 5$ captions per rendered image using nucleus sampling [96]. By generating multiple captions, we increase the likelihood of generating correct details (e.g. "black and yellow toy bomb" in Figure 2 view $M$ caption 1). Incorrect captions, such as "scissors" in Figure 2 view $M$ caption $N$, can be filtered in later stages. To generate captions containing fine-grained geometry details (in our ABO experiments), we employ a two-stage question-answering instead of captioning. The first stage

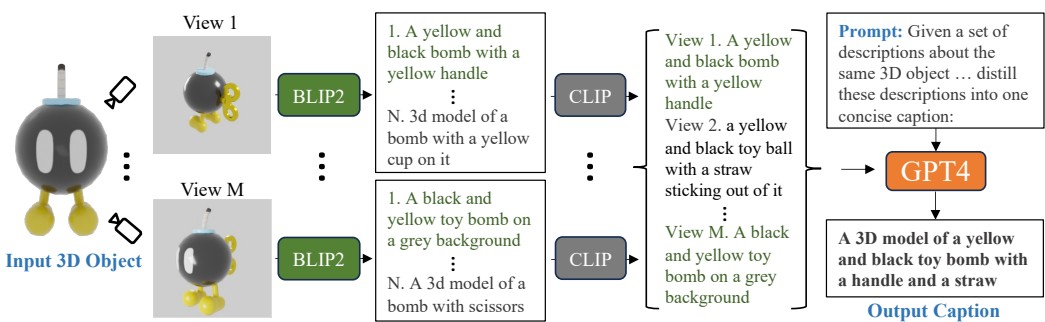

Figure 2: **Overview of Cap3D**. Left to Right: (1) Render 3D objects from $M = 8$ camera angles to capture object details (2) Generate $N = 5$ image captions per rendered image using BLIP2; (3) Select one caption for each image based on its similarity to the image encoding using CLIP; (4) Use GPT4 to consolidate all selected captions into a final, summary of the object.

generates one answer to a prompt asking what object is pictured. The answered object is passed into a second prompt, which asks its structure and geometry, and generates 5 answers.

**Caption Selection:** While BLIP2 often generates high-quality captions, it is not uncommon for samples to contain mistakes, particularly in non-forward facing views such as "yellow cup", in Figure 2 view 1, caption $N$. To reduce the frequency of mistakes, we compute CLIP [16] ViT-B/32 [54] encodings from each of 5 captions and the associated image, and select the caption maximizing cosine similarity. CLIP tends to select good captions for each view, e.g. Figure 2: view 1, BLIP2 caption 1 and view $M$, caption 1. CLIP is complementary to BLIP2 as not only does it have different training details and architecture, but it trains on different data. While BLIP2 is trained upon COCO [31], Visual Genome [32], CC3M [11], CC12M [12], SBU [33] and LAION400M [13]; CLIP is trained upon a dataset of 400M images based on frequent text occurrence in Wikipedia.

**Caption Consolidation:** Accumulating information across viewpoints to form a complete picture of 3D objects is challenging, but crucial. We find prompting of GPT4 [63] to summarize the $M$ captions results in good parsing of the details across captions. By applying GPT4 as the final summary step, it can both include significant details and remove unlikely ones. For example, the final caption in Figure 2 filters the incorrect information, from view 2, "toy ball", while keeping key details, including "handle" and "straw". The alternative order of GPT4 followed by CLIP would result in (1) GPT4 having to make sense of more incorrect input details and (2) CLIP simply selecting between aggregate captions instead of being able to error-correct small mistakes. The effectiveness of introducing GPT4 is verified in ablations (Table 3).

## 3.2 Ethical Filtering

Captions generated and images rendered by Cap3D enhance the identification and mitigation of legal and ethical issues associated with large-scale 3D object datasets, including identifiable information and NSFW content.

We manage two datasets: Objaverse and ABO. In Objaverse, our main responsibility involves dealing with artist-created assets. These can include identifiable elements such as human face scans and NSFW objects. Objaverse contains approximately 800k objects, which makes the manual verification of each asset impractical. The ABO dataset, on the other hand, is smaller and mostly consists of furniture. We manually ensure the ethical integrity of this dataset.

We exclude objects that lack sufficient camera information for rendering, leaving us with 785k objects. These objects encompass a mix of non-commercial and commercial licenses, such as CC BY-NC-SA, CC BY-NC, CC BY, CC BY-SA, and CC0. We then applied ethical filtering exclusively to objects with commercial-friendly licenses (i.e., CC BY, CC BY-SA, and CC0), resulting in a count of 680k objects.

We next follow prior work [10] and use a face detector [97] and NSFW classifier [98, 99] on forward-facing object renders and filter detected objects with score $>= 0.9$. The face detector filters out 18.6k

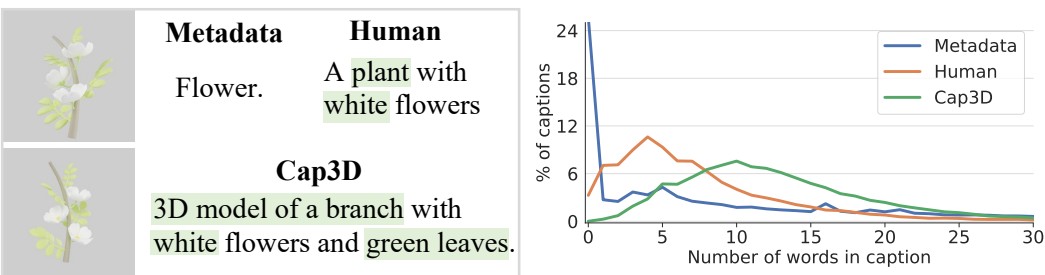

Figure 3: **Objaverse Caption Comparison.** Human captions and Internet metadata frequently contain limited detail. Cap3D captions typically have longer length and more detail. Vocabular size comparisons among Cap3D, BLIP2, and human captions are included in Appendix E.

objects, and the NSFW classifier filters out 217 objects. Text is also carefully processed. Our final captions are the output of GPT4, which has been trained to filter out inappropriate or harmful content [63]. We run a standard blocklist [100] on its output, removing any object-caption pairs including blocked words. This filters out 226 objects. After all the filtering, we are left with 661k objects in the Objaverse dataset. We manually estimate detection precision and recall in Table 2. To summarize, our process detects over 19k objects, of which a nontrivial amount is accurately removed. We estimate roughly 1k face and less than 1k NSFW are missed, using a conservative standard (e.g. missed faces are typically sports cards), resulting in a 660k subset.

Table 2: **Ethical Filtering Analysis.** We manually detect faces and NSFW content to validate automated filtering. 16 of 17 missed face detections were sports cards.

| | Detected | Precision | | Missed dets. | |
|---|---|---|---|---|---|
| | (Filtered) | 5k | (%) | 10k | 680k |
| Faces | 18.6k | 790 | 16% | 17 | ≈1k |
| NSFW | 217 | 102 | 47% | 12 | <1k |
| Language † | 226 | – | – | – | – |

†: *String match filtering is deterministic.*

## 4 Dataset

We collect captions in two distinct settings: Objaverse, a large and varied dataset of artist-created 3D assets; and ABO, a small dataset of real products, typically furniture.

### 4.1 Objaverse Captions

Objaverse [9] features roughly 800k 3D object assets across 21k classes designed by over 100k artists. It is of significantly larger scale than prior work; the paper shows this size enables more diversity by generative 3D models trained upon it. It is released under the ODC-By 1.0 license, permitting subsequent researchers to curate new data from it. Metadata is paired with many assets, however as seen in Figure 3 (right), metadata caption length is frequently short or empty. We collect two caption datasets on Objaverse. First, an automated set of one caption for each of 785k objects using Cap3D (a total of 785k captions). Second, a crowdsourced set of 41.4k captions spanning 39.7k objects for evaluating generated captions. Captions are collected using thehive.ai, a crowdsourced platform similar to AMT. Workers are given instructions with gold-standard sample captions, see the same 8 views as models during captioning, and are routinely monitored. Poor captioning performance results in a ban and deletion of the worker's captions. Crowdsourced captions are also filtered using the blocklist in § 3.2. Figure 3 (left) shows human captions provide more detail than metadata, but automated captions tend to be most descriptive.

### 4.2 ABO Geometry Captions

ABO [35] is a collection of 3D models of Amazon products and is primarily furniture. ABO serves as an important contrast to Objaverse as it consists of a small number of classes varying primarily in geometry. Captioning, therefore, needs to focus more on structure as opposed to semantic category. To emphasize this focus, we consider the task of captioning the geometric structure of objects without color or texture (seen in the bottom right of Figure 1). Like Objaverse, ABO contains metadata that is

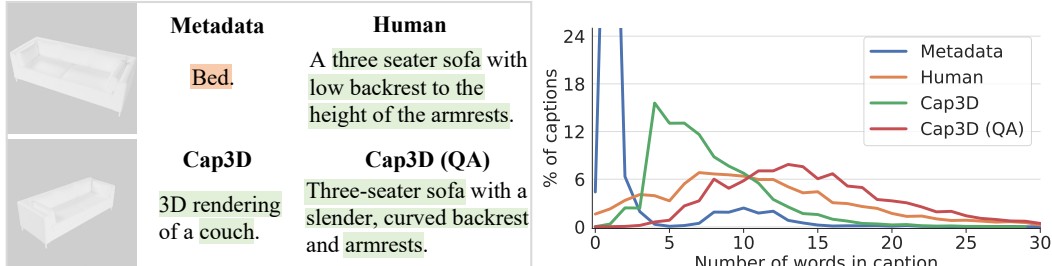

Figure 4: **ABO Automated Geometric Description**. **Left:** Human descriptions provide more detailed geometry than automated captions. With careful prompting, *Cap3D (QA)* can match human-level detail. **Right:** The high peak of Metadata is cropped, which otherwise obscures other curves.

typically quite short (Table 4), resulting in limited detail. We collect three sets of captions on the 6.4k ABO splits of [78]: crowdsourced (a total of 17.2k captions), captions generated by Cap3D (a total of 6.4k captions), and captions generated by Cap3D (QA) which uses the two-stage prompt captioning (a total of 6.4k captions). Crowdsourced captions follow similar detail to Objaverse with the exception instructions and examples are focused on geometric structure. We compare alternatives in Figure 4. In contrast to Objaverse, human geometric descriptions on ABO are more detailed than captioning. With prompting (QA), the Cap3D pipeline can rival human descriptions.

## 5   Experiments

In this section, we first validate the quality of Cap3D captions against metadata and human-authored captions on both Objaverse and ABO. To verify Cap3D captions are helpful in practice, we next compare text-to-3D models finetuned on both human-authored captions and Cap3D (using the same >30k set as crowdsourced captions). Finally, we evaluate state-of-the-art text-to-3D models on our captions at scale to measure if finetuning on our captions can improve performance.

### 5.1   3D Captioning on Objaverse

**Dataset.** We evaluate caption quality on three subsets of Objaverse: (1) a random set of 22k objects containing a human caption, (2) a random split of 5k objects containing a human caption, and (3) a random 5k split across the entire dataset.

**Baselines.** In data splits (1) and (2), we compare the caption generated by Cap3D with human-authored annotations, *Human*, and existing Objaverse metadata, *Metadata*, described in § 4.1. Split (1) is used for A/B testing of *Cap3D* vs. *Human*, as shown in Table 1, at scale. Collecting A/B comparison is expensive, so we compute more extensive experiments on the smaller set (2) in Table 3.

In data split (3), we ablate the main components of Cap3D into *BLIP2* and *+GPT4*. *BLIP2* uses only the image captioning component of our method, taking a front-view rendering and producing a single output caption. *+GPT4* uses the same image captioning process of our method, producing 5 captions for each of 8 views. However, instead of using CLIP to filter 5 captions from each view, it directly summarizes all 40 captions into a final caption.

**Metrics.** Our primary metric is human judgment A/B tests, where we ask workers to select between two captions on a scale of 1-5, where 3 is a tie. Workers are carefully monitored and each comparison has at least 10k observations across 5k objects. We report mean score, along with the percent each method is preferred (i.e. scores a 4 or 5). We use automated metrics CLIPScore [16, 62], the cosine similarity of CLIP encodings with input images; and ViLT Image and Text Retrieval, which ranks likely image-text pairs, from which one computes precision.

We emphasize CLIPScore is not our primary metric since our captioning model utilizes CLIP. BLIP2 utilizes ViT-L/14 and ViT-g/14, while our filtering uses ViT-B/32, so following previous work [85] we compute CLIP score using a different model to reduce bias (ViT-B/16). However, we report it as it

Table 3: **Objaverse Captions Evaluations**. *Cap3D* outperforms *human* and *Metadata*; *BLIP2*, *GPT4*, and *CLIP* are all important to performance. We report 95% confidence interval and use 5k objects.

| Method | User A/B Study vs. Cap3D | | | CLIP Score | ViLT Img Retr. | | ViLT Text Retr. | |
| | Score (1-5) | Win % | Lose % | | R@5 | R@10 | R@5 | R@10 |
|---|---|---|---|---|---|---|---|---|
| Metadata | 1.74±0.026 | 10.7 ± 0.7 | 83.8 ± 0.8 | 66.8 | 4.3 | 6.3 | 6.1 | 8.5 |
| Human | 2.86±0.026 | 37.0±1.0 | 46.1±1.0 | 72.5 | 21.2 | 29.0 | 18.5 | 24.9 |
| Cap3D | - | - | - | **88.4** | **35.7** | **46.3** | **34.7** | **44.2** |
| BLIP2 | 2.87± 0.019 | 41.0± 0.7 | 50.6± 0.7 | 83.1 | 24.7 | 32.3 | 21.9 | 29.3 |
| + GPT4 | 2.94± 0.015 | 35.2± 0.6 | 40.8± 0.6 | 86.3 | **31.9** | 39.9 | 30.2 | 38.4 |
| + CLIP (Cap3D) | - | - | - | **86.9** | 31.1 | **40.2** | **30.3** | **38.6** |

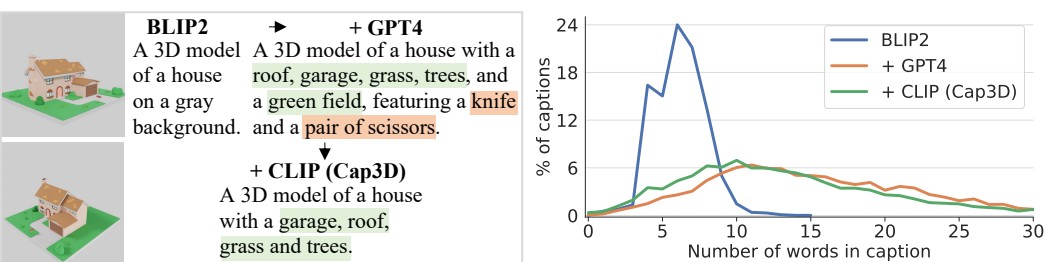

Figure 5: **Objaverse Caption Ablations**. GPT produces longer and more detailed captions than BLIP2; CLIP tends to prune incorrect details and reduces length slightly.

has shown a higher correlation with human judgments than other automated metrics [62]. ViLT [101] is trained on different data and is a different architecture than CLIP, providing an orthogonal metric.

**Results.** We report large scale A/B testing (1) against *Human* in Table 1, which shows Cap3D is better across metrics, with high confidence. The top three rows of Table 3 use the smaller human-captioned split (2), and demonstrate Cap3D's superior performance over Objaverse metadata and human-authored captions across A/B studies and automated metrics. The bottom three rows of Table 3, studied across a random split of the full dataset (3), reveal that while *BLIP2* is effective, incorporating multiple views with *+GPT4* enhances performance. As shown in Figure 5, GPT4 adds detail by consolidating view-specific information. Filtering using *+CLIP (Cap3D)* mitigates false details by purging subpar captions from GPT input. In addition to reducing errors, utilizing CLIP also reduces GPT input captions from 40 to 8, effectively decreasing token numbers and facilitating a cost reduction from $15.33 to $4.18.

## 5.2 Geometry 3D Captioning on ABO

**Dataset.** We evaluate geometric captioning on a 6.4k object split from ABO [35, 78], comparing Cap3D captions for each object against a maximum of two human-authored ones. To emphasize geometric focus, images used for model input and human assessment are texture-free and colorless.

**Baselines and Metrics.** We use two automated variants from §3.1: *Cap3D* and *Cap3D (QA)*, which uses a two-stage prompt captioning to ask more about the input 3D geometry; and compare to crowdsourced human descriptions, *Human*, detailed in §4.1, and ABO metadata, *Meta*.

Our primary metric of comparison is similar human A/B testing to §5.1, since automated metrics such as CLIPScore do not accurately represent the distance between fine-grained captions and images as shown in [78].

**Results.** In stark contrast to Objaverse, *Human* captions beat automated (*Cap3D*) in Table 4. Automated captions alone contain little geometric detail (e.g., Figure 4), making *Cap3D* unsuited for this setting. However, by using the two-stage prompt engineering, *Cap3D (QA)* is preferred to *Human*. Shown in Figure 4, *Cap3D (QA)* produces significant fine-grained geometric detail as well as longer captions in general. In contrast, *Metadata* is clearly the weakest baseline.

Table 4: **ABO Fine-Grained Geometry Captions**. Cap3D (QA) performs best; crowd-sourced beats captioning alone.

| Method | A/B Score (1-5) | A/B Win % | A/B Lose % |
|---|---|---|---|
| Human v. Cap3D | 3.09±0.02 | 47.3±1% | 41.4±1% |
| Cap3D(QA) v. Human | 3.08±0.02 | 50.2±1% | 44.0±1% |
| Cap3D(QA) v. Cap3D | 3.27±0.02 | 56.0±1% | 37.4±1% |
| Cap3D(QA) v. Meta | 4.27±0.02 | 88.2±1% | 10.0±1% |

Table 5: **Text-to-3D: Human Captions**. Cap3D captions are better than human on the 30k set. Finetuning on Cap3D full set performs best.

| | Finetune Dataset | FID↓ | CLIP Score | CLIP R-Precision (2k) | | |
|---|---|---|---|---|---|---|
| | | | | R@1 | R@5 | R@10 |
| Point·E | Pretrained | 36.1 | 72.4 | 6.0 | 16.2 | 22.4 |
| | 30k (Human) | 34.6 | 74.4 | 8.2 | 21.3 | 29.1 |
| | 30k (Cap3D) | 33.7 | 75.0 | 10.4 | 24.3 | 32.1 |
| | 350k (Cap3D) | **32.8** | **75.6** | **12.4** | **28.1** | **36.9** |
| Shap·E | Pretrained | 37.2 | **80.4** | **20.3** | **39.7** | **48.7** |
| | 30k (Human) | 36.0 | 79.6 | 18.6 | 36.3 | 45.3 |
| | 30k (Cap3D) | 37.2 | 79.4 | 19.1 | 37.5 | 46.1 |
| | 350k (Cap3D) | **35.5** | 79.1 | 20.0 | 38.8 | 47.3 |

## 5.3 Large-Scale Text-to-3D Generation

**Dataset.** We evaluate text-to-3D generation on three subsets of Objaverse: (1) a 30k split of objects containing human-authored captions, to measure if finetuning on Cap3D captions outperform human-authored ones; (2) a 350k split of Objaverse objects paired with Cap3D captions, for finetuning state-of-the-art text-to-3D methods – obtaining high-density point cloud and latent codes to finetune Point·E and Shap·E for all 785k objects is prohibitively expensive (20k GPU days); and (3) a 300 object split for optimization-based baselines, which typically take >30 mins per object to optimize. Pretrained and Finetuned models are evaluated on 8 views across a held-out test set of 2k objects.

**Methods.** We consider several recent SOTA methods in three general categories: text-to-3D diffusion, cascaded text-to-image then image-to-3D diffusion, and optimization-based. We use the direct text-to-3D variant of *Point·E* [89], as well as two variants of *Shap·E* [90]: *STF* [102] and *NeRF* [103]. We use *Stable Diffusion* cascaded with *Point·E (Im-to-3D)*, adapting *ControlNet* [64] and *LoRA* [104] for Stable Diffusion finetuning. We use optimization-based baselines *DreamField* [85], the publicly available implementation of *DreamFusion* [3], Stable DreamFusion [105]; and *3DFuse* [106], using their implementation based on Karlo [24, 107].

**Metrics.** We use standard metrics from prior work [3, 85, 89, 90] to evaluate. Primarily, these are CLIP Score and CLIP R-Precision. CLIP R-Precision ranks a rendered image against all text pairs in the test set by CLIP cosine similarity, and computes precision upon true text-image correspondence. Since we have ground truth images, we calculate the FID [108] of 3D rendered images against ground truth images, as well as assess CLIP Score on these reference images. We also use ViLT Retrieval R-Precision, used in 5.1, which has the same evaluation procedure as CLIP R-Precision with a different model.

**Results.** Table 5 lists the results of finetuning using human-authored and Cap3D captions. Point·E improves after finetuning upon human captions. However, performance is further improved using our captions on the same dataset; and improved most by training upon the full dataset. This result strongly defends Cap3D captioning at scale. Shap·E does not improve on CLIP metrics after finetuning in any dataset, but performs the least bad on the full dataset using our captions; and FID improves most.

Table 6 presents results from several state-of-the-art pretrained and finetuned models using Cap3D-generated captions. The models finetuned on our captions generally outperform pretrained models under the FID metric. For CLIP-related metrics, the finetuned models of *Point·E (Text-to-3D)* and *StableDiffusion + Point·E (Im-to-3D)* also beat their pretrained counterparts. Point·E and Stable Diffusion have been trained on massive datasets, so improvement from finetuning is strong evidence Cap3D captions are effective. The observed downturns in *Shap·E* could be attributed to at least two factors. First, our replication of their privately-available train code is unstable, often resulting in NaN loss during finetuning. We restart from earlier checkpoints upon crashing, but the result alone is concerning. Second, we exclusively finetune the diffusion model in Shap·E's two-stage approach.

Qualitative results in Figure 6 validate quantitative findings. *Point·E* and *Stable Diffusion* baselines show large improvements from finetuning, while *Shap·E* can better fit the Objaverse data distribution (corresponding to improved FID).

Table 6: **Text-to-3D on Objaverse**. Finetuning improves FID over pretrained performance across models. CLIP metrics of *Stable Diffusion* increase; CLIP metrics of *Point·E* increase significantly.

| | Pretrained | | | | | Finetuned on Cap3D | | | | |
| | FID↓ | CLIP Score | CLIP R-Precision (2k) | | | FID↓ | CLIP Score | CLIP R-Precision (2k) | | |
| | | | R@1 | R@5 | R@10 | | | R@1 | R@5 | R@10 |
|---|---|---|---|---|---|---|---|---|---|---|
| Ground Truth Images | - | 81.6 | 32.7 | 55.1 | 64.3 | - | 81.6 | 32.7 | 55.1 | 64.3 |
| Point·E (Text-to-3D) [89] | 36.1 | 72.4 | 6.0 | 16.2 | 22.4 | **32.8** | **75.6** | **12.4** | **28.1** | **36.9** |
| S. Diff. [22] (CNet) [64]+ [89](Im-to-3D) | 54.7 | 73.6 | 11.0 | 23.4 | 30.0 | **53.3** | **74.6** | **12.4** | **26.2** | **33.8** |
| S. Diff. [22] (LoRA) [104]+ [89](Im-to-3D) | 54.7 | 73.6 | 11.0 | 23.4 | 30.0 | **53.7** | **74.4** | **11.6** | **24.6** | **31.4** |
| Shap·E [90] (STF) [102] | 37.2 | **80.4** | **20.3** | **39.7** | **48.7** | **35.5** | 79.1 | 20.0 | 38.8 | 47.3 |
| Shap·E [90] (NeRF) [103] | 48.7 | **79.4** | **19.0** | **37.7** | **46.8** | **48.2** | 78.1 | 18.3 | 35.1 | 43.5 |

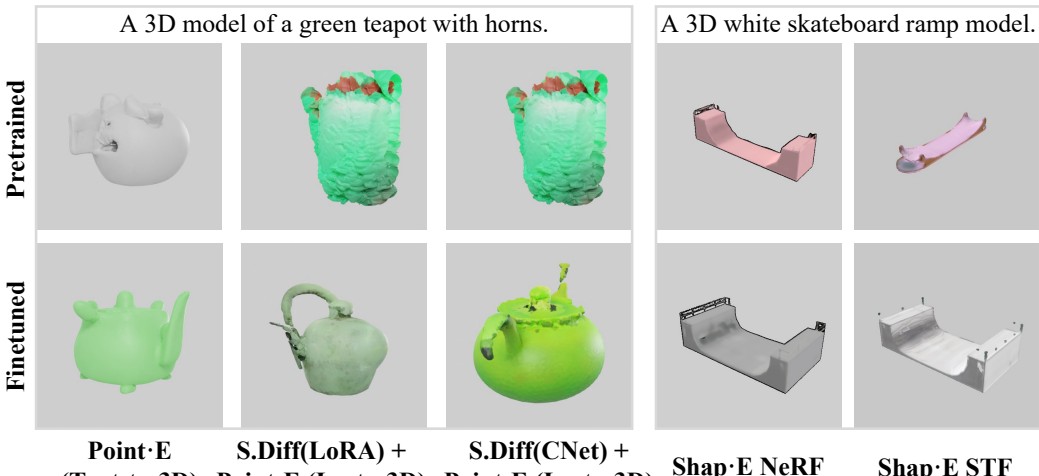

| Pretrained | A 3D model of a green teapot with horns. | | | A 3D white skateboard ramp model. | |
| Finetuned | | | | | |
| | **Point·E (Text-to-3D)** | **S.Diff(LoRA) + Point·E (Im-to-3D)** | **S.Diff(CNet) + Point·E (Im-to-3D)** | **Shap·E NeRF** | **Shap·E STF** |

Figure 6: **Text-to-3D results.** Finetuning on Cap3D captions can significantly improve results. Additional examples are available in Appendix C.

Optimization baselines, shown in Table 7, perform very well upon CLIP-based metrics, consistent with prior work [90]. In fact, *DreamField* outperforms ground truth images in CLIP metrics. This demonstrates *DreamField* overfits to the CLIP metric, which is the standard protocol for text-to-3D evaluation. We propose to also consider ViLT precision (see §5.1). This helps mitigate the bias of CLIP, though *DreamField* performance on this metric is still strong.

Table 7: **Text-to-3D: Optimization Baselines**. Overfitting via CLIP leads to higher CLIP-based scores than ground truth; ViLT score is more fair.

| | FID↓ | CLIP | | | ViLT | |
| | | Score | R@1 | R@5 | R@1 | R@5 |
|---|---|---|---|---|---|---|
| True Images | - | 83.2 | 53.2 | 77.8 | 41.3 | 69.0 |
| D. Field [85] | 106.1 | **83.7** | **61.8** | **83.6** | **32.3** | **56.0** |
| D. Fusion [3] | 127.8 | 72.4 | 28.4 | 46.1 | 23.7 | 45.3 |
| 3DFuse [106] | **91.1** | 77.0 | 38.6 | 58.5 | 26.3 | 53.0 |

## 6 Limitations and Future Works

As described in §3, Cap3D consists of four steps: (1) 3D objects rendering; (2) captioning via BLIP2; (3) filtering captions via CLIP; (4) consolidate multiview information via GPT4. To effectively capture comprehensive information through 2D renderings, cameras are strategically placed above or below objects. However, this occasionally results in unconventional 2D views, making BLIP2 susceptible to errors that CLIP fails to rectify. This, in turn, hampers GPT4's ability to merge variegated information across views, culminating in vague and verbose descriptions, as illustrated in Figure 7. The system also falters with certain complex indoor 3D scans, as depicted in Figure 8, thus requiring more robust image-captioning models [30] and potentially benefiting from additional view incorporations beyond the current eight.

Our method's provision of extensive 3D-Caption pairs for Objaverse [9] could foster the advancement of 3D-LLM models [109, 110], facilitating 3D-caption centric tasks like captioning, dialog, and language-based navigation. Additionally, the geometric descriptions generated for ABO [35] enable

compositional structure analysis of fine-grained 3D objects [111, 112]. Our developed method assists in scaling up of 3D-text pairs for expansive 3D datasets [113].

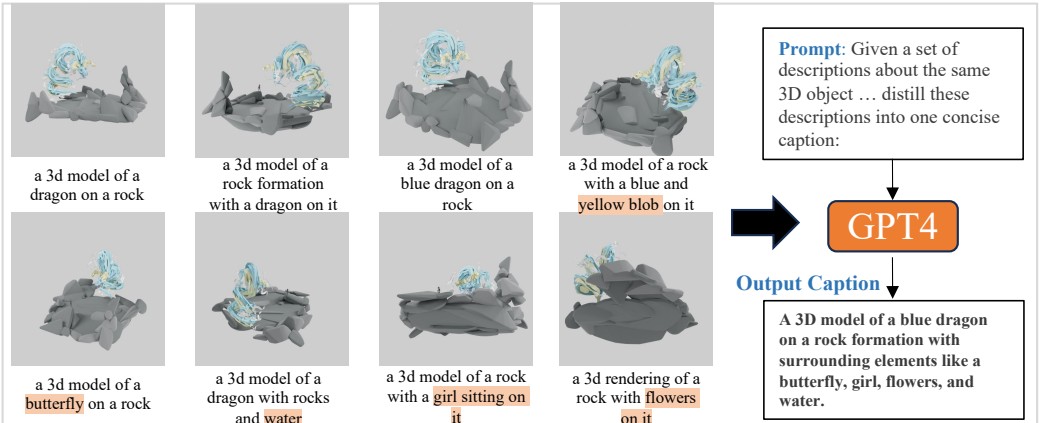

Figure 7: An failed case. The caption under each rendered image are generated by BLIP2 + filtered by CLIP. The inaccurate content are highlighted with colors. GPT4 + CLIP cannot fix the error generated by BLIP2 and result in a fuzzy description.

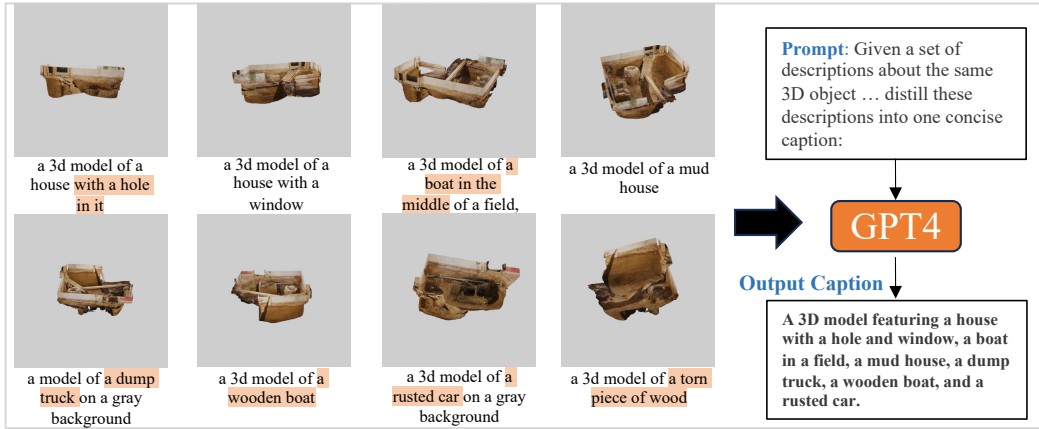

Figure 8: An failed case. The caption under each rendered image are generated by BLIP2 + filtered by CLIP. The inaccurate content are highlighted with colors. The various views contain inaccurate information. The associated details, roughly described, fail to accurately depict the indoor scene.

## 7  Conclusion

In this work, we collect (1) 3D object captions at scale, creating the largest publicly available high-quality 3D-text by an order of magnitude. To do so we propose Cap3D, an automated pipeline leveraging several models pretrained on large datasets, and show design choices are important to performance. In addition, we collect (2) a dataset of geometric captions upon fine-grained 3D objects. This helps analyze shortcomings of automated captioning and study the potential of question answering, while yielding geometric descriptions for 3D assets of real objects paired with real images. These datasets serve as benchmarks for text-to-3D tasks (1) at scale and (2) in geometric detail.

## Acknowledgments and Disclosure of Funding

This work is supported by two grants from LG AI Research and Grant #1453651 from NSF. We greatly thank Kaiyi Li for his technical support. We thank Mohamed EI Banani, Karan Desai, and Ang Cao for their helpful discussions. Thanks Matt Deitke for helping with Objaverse-related questions. Thanks to Haochen Wang for helping notice some incorrect rendering.

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

# Appendix A  Price Breakdown Details

This section provides our details computation for Table 1. Using a single A40 GPU, BLIP2 runs at $\sim 2700$ iterations per hour, enabling it to process around $\sim 337.5$ objects hourly given the eight-run requirement for generating captions for 8 rendering views. This translates to about 2.96 hours to process 1k objects, costing $2.96 \times \$1.28 = \$3.79$ with the rate $\$1.28/hr$ on the cloud platform, CoreWeave. On the same A40 GPU, CLIP operates at $\sim 27000$ iterations per hour, incurring a cost of \$0.38. Importantly, utilizing eight A40s costs the same as using one, due to the parallel processing capacity across multiple GPUs for multiple rendering views.

We compute our GPT4 cost by averaging input token numbers, as OpenAI GPT4 API (8k context) costs $0.03/1k$ tokens, Our input prompt is: "Given a set of descriptions about the same 3D object, distill these descriptions into one concise caption. The descriptions are as follows: 'captions'. Avoid describing background, surface, and posture. The caption should be:", which consists of (1) text prompt and (2) captions generated by BLIP2 or BLIP2 + CLIP. Without CLIP's filtering, our input prompt contains 40 captions which have $\sim 511.1$ tokens on average, cost $511.1/1000 \times 0.03 \times 1000 = \$15.33$ for $1k$ objects. With CLIP, our input prompt contains 8 captions which have $\sim 139.3$ tokens on average, cost $139.3/1000 \times 0.03 \times 1000 = \$4.18$ for $1k$ objects.

The average cost per $1k$ objects for human-authored annotation is computed as the average expenditure on the crowdsourcing platform, Hive. The human annotation speed is computed by averaging the annotation progress across our whole annotation process.

We do not report the average cost of Cap3D (QA) in the main paper, as we only use it on ABO. For completeness, we report it here. The one distinction is BLIP2 is run twice instead of once for the two-stage question answering (QA). The cost of BLIP2 thus doubles, from \$3.79 to \$7.58; and total cost increases from \$8.35 to \$12.14 per 1k objects.

# Appendix B  Additional 3D Captioning Results

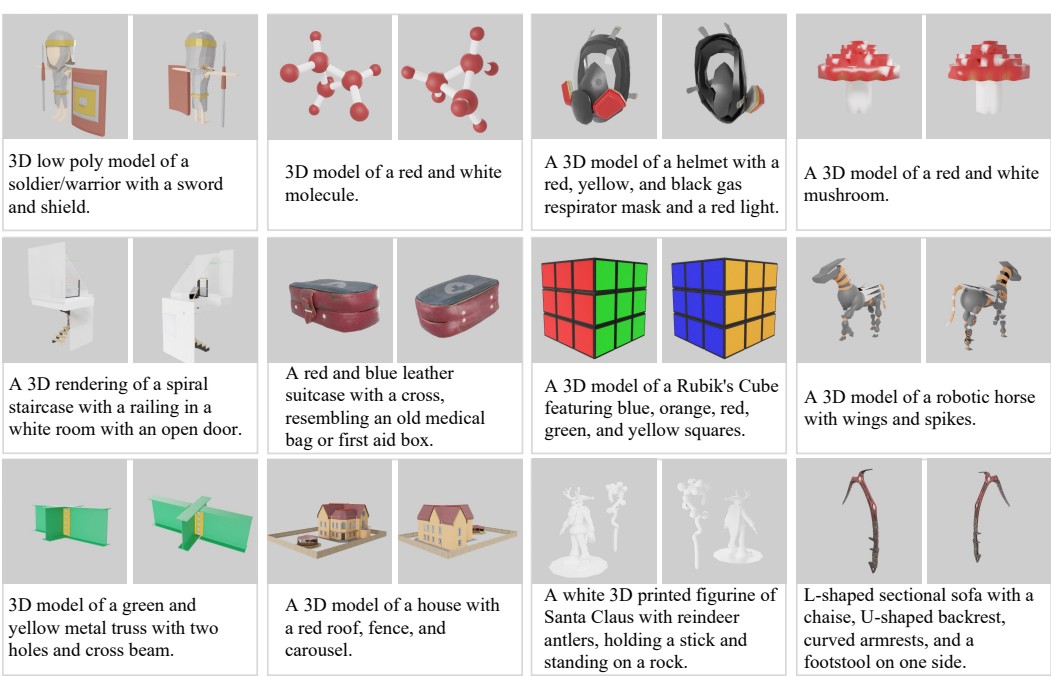

| | |
|---|---|
| 3D low poly model of a soldier/warrior with a sword and shield. | 3D model of a red and white molecule. |
| A 3D model of a helmet with a red, yellow, and black gas respirator mask and a red light. | A 3D model of a red and white mushroom. |
| A 3D rendering of a spiral staircase with a railing in a white room with an open door. | A red and blue leather suitcase with a cross, resembling an old medical bag or first aid box. |
| A 3D model of a Rubik's Cube featuring blue, orange, red, green, and yellow squares. | A 3D model of a robotic horse with wings and spikes. |
| 3D model of a green and yellow metal truss with two holes and cross beam. | A 3D model of a house with a red roof, fence, and carousel. |
| A white 3D printed figurine of Santa Claus with reindeer antlers, holding a stick and standing on a rock. | L-shaped sectional sofa with a chaise, U-shaped backrest, curved armrests, and a footstool on one side. |

Figure 9: Random 3D captioning examples generated by Cap3D. Two views of 3D objects (Objaverse [9]) are shown here, Cap3D uses eight.

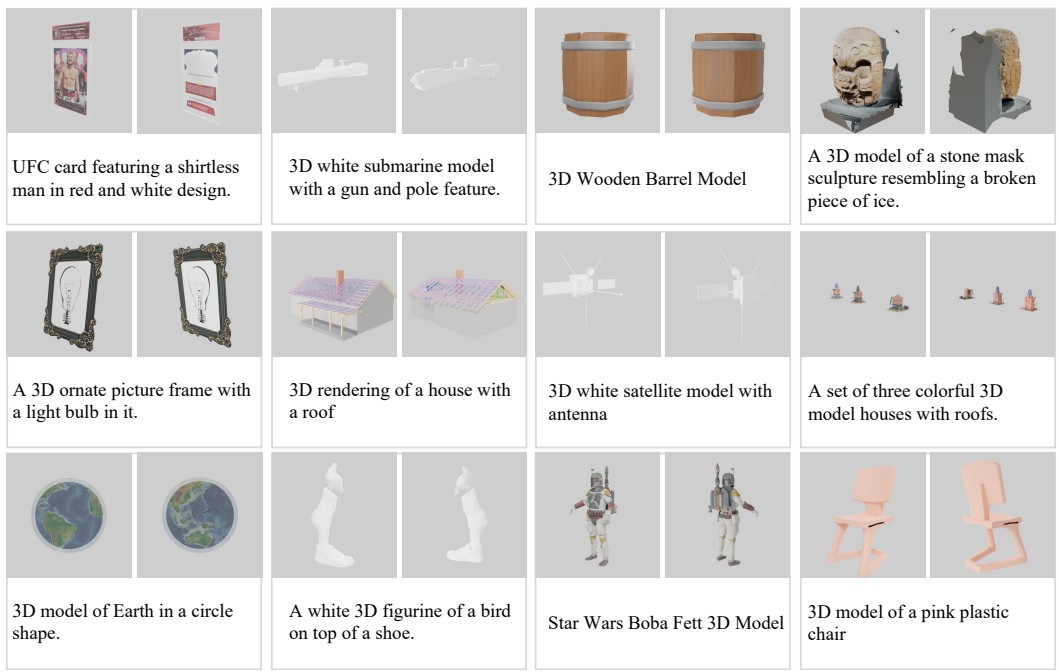

Figure 10: Random 3D captioning examples generated by Cap3D. Two views of 3D objects (Objaverse [9]) are shown here, Cap3D uses eight.

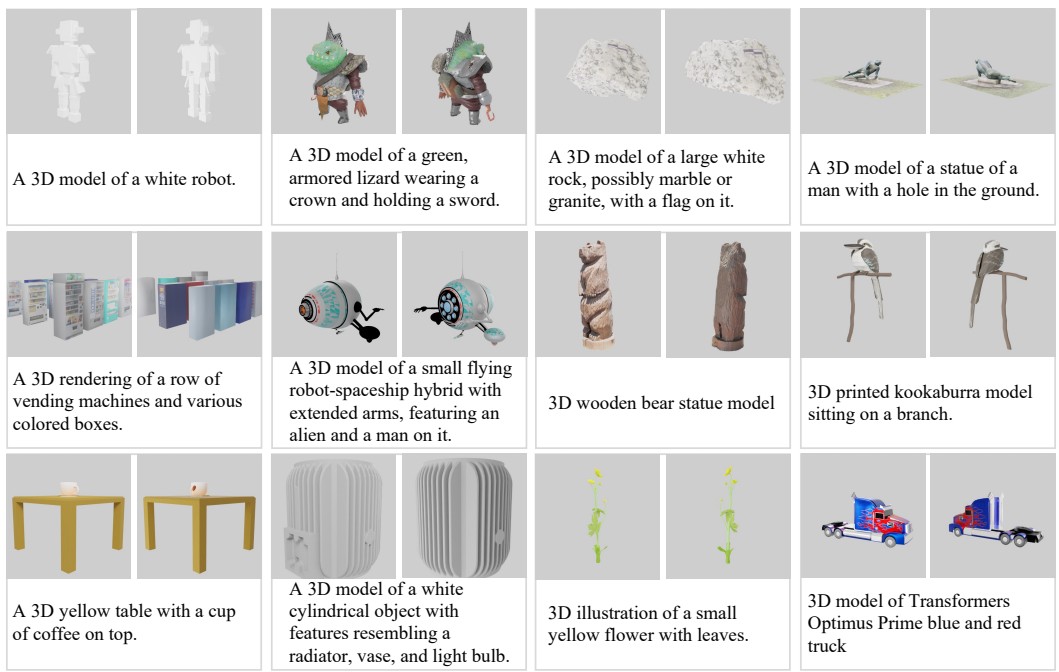

Figure 11: Random 3D captioning examples generated by Cap3D. Two views of 3D objects (Objaverse [9]) are shown here, Cap3D uses eight.

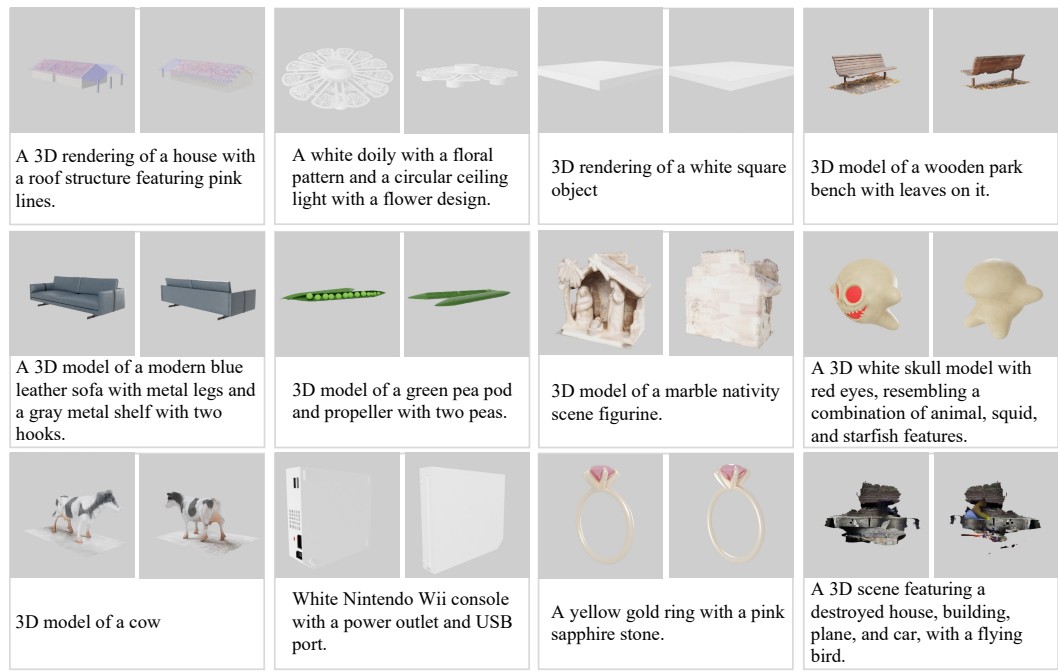

Figure 12: Random 3D captioning examples generated by Cap3D. Two views of 3D objects (Obja-verse [9]) are shown here, Cap3D uses eight.

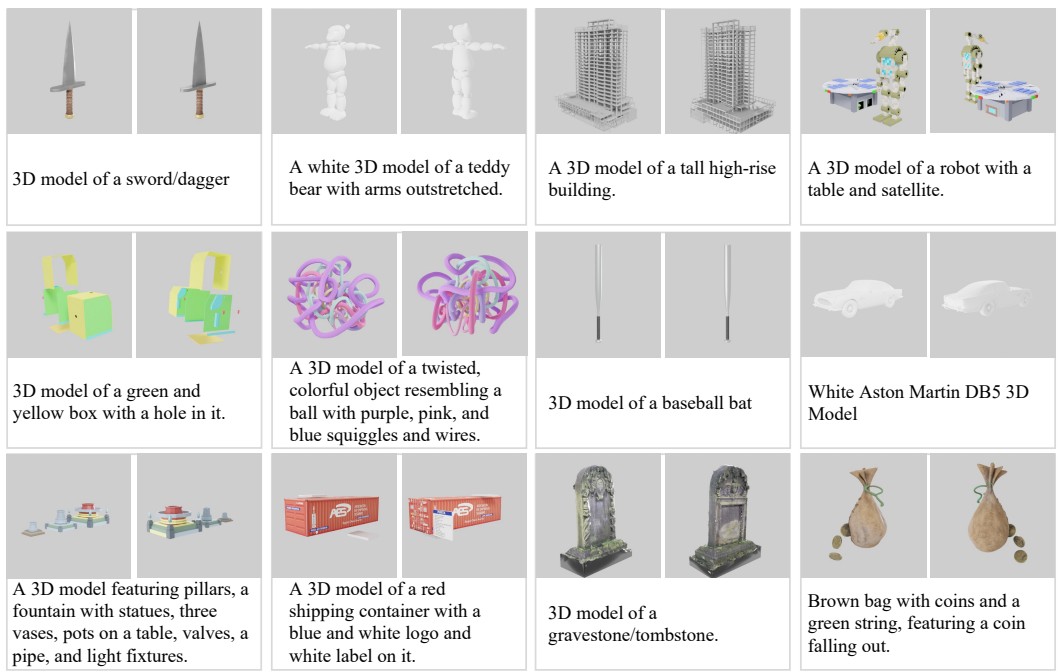

Figure 13: Random 3D captioning examples generated by Cap3D. Two views of 3D objects (Obja-verse [9]) are shown here, Cap3D uses eight.

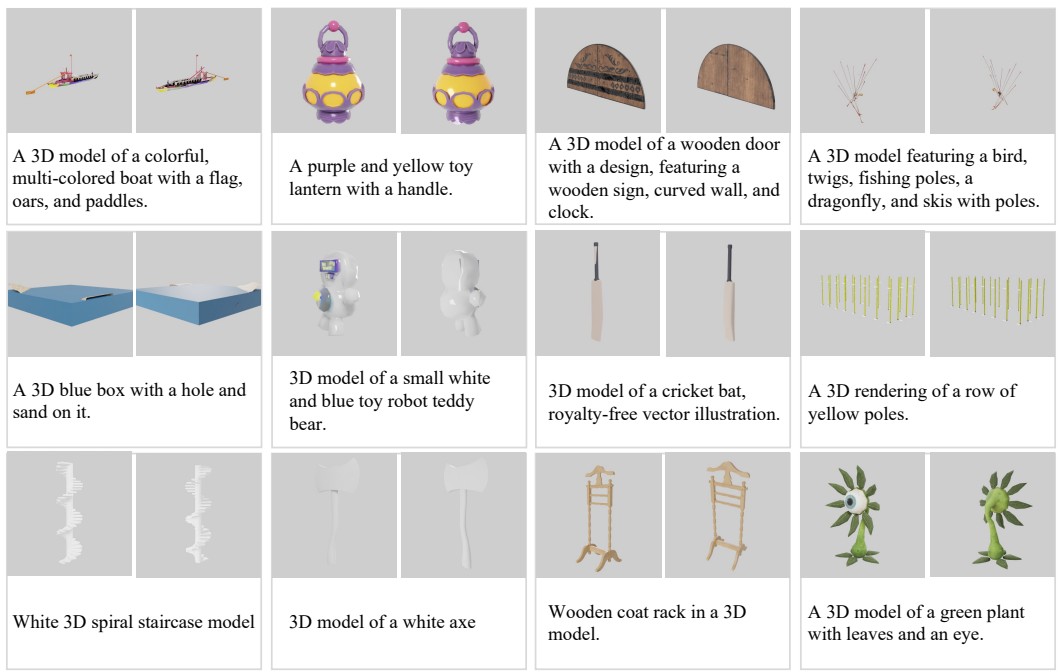

Figure 14: Random 3D captioning examples generated by Cap3D. Two views of 3D objects (Objaverse [9]) are shown here, Cap3D uses eight.

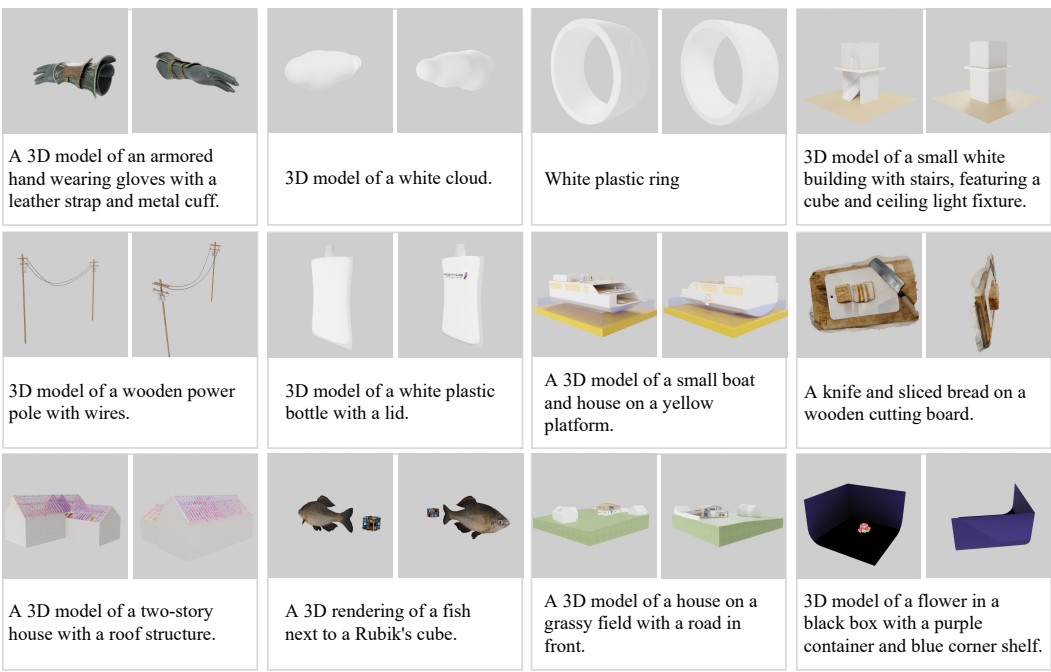

Figure 15: Random 3D captioning examples generated by Cap3D. Two views of 3D objects (Objaverse [9]) are shown here, Cap3D uses eight.

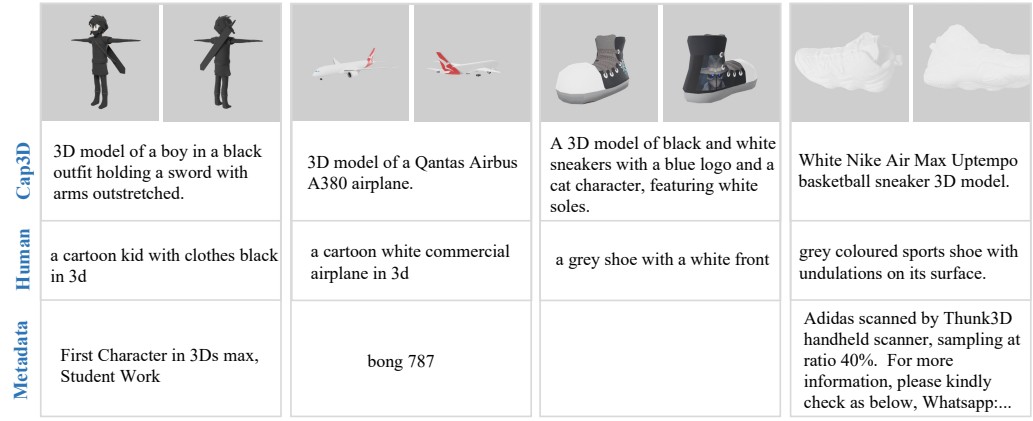

Figure 16: Comparative Analysis: Cap3D Generated Caption vs Human-Annotated Caption vs Objaverse Metadata [9]. Two views of 3D objects are shown here, Cap3D and human use eight.

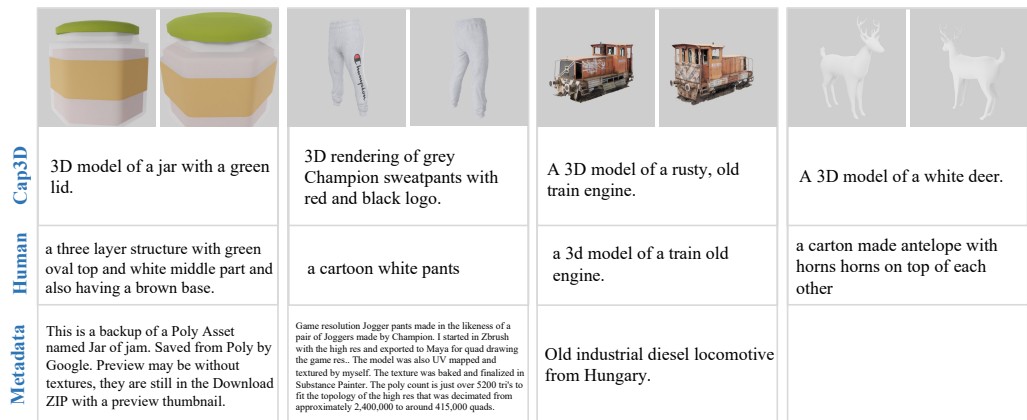

Figure 17: Comparative Analysis: Cap3D Generated Caption vs Human-Annotated Caption vs Objaverse Metadata [9]. Two views of 3D objects are shown here, Cap3D and human use eight.

## Appendix C  Additional Text-to-3D Results

In this section, we provide several text-to-3D results for all of our compared methods. We include Shap·E and Point·E pretrained models and the models finetuned on our data, as well as optimization baselines, including DreamFusion, DreamField, and 3D Fuse.

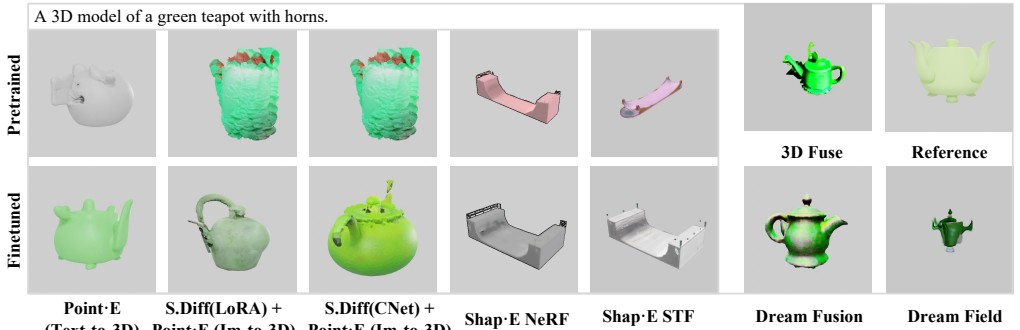

Figure 18: Text-to-3D results. The top text prompt and "Reference" are from our test set. We fine-tune the left 5-column methods on Cap3D-generated captions. The detailed setting and methods are described in §5.3.

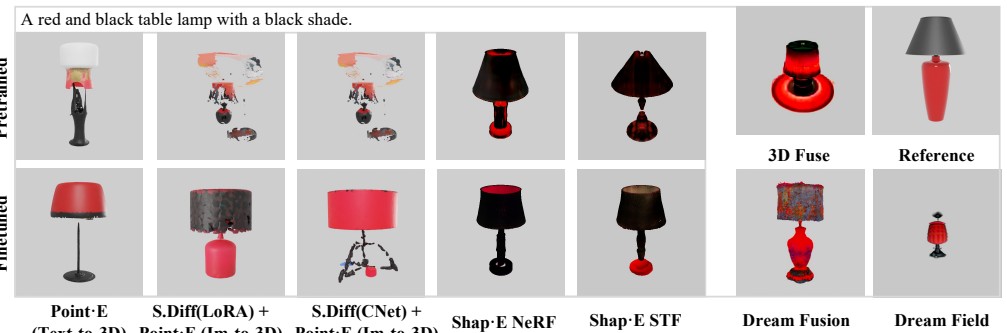

Figure 19: Text-to-3D results. The top text prompt and "Reference" are from our test set. We fine-tune the left 5-column methods on Cap3D-generated captions. The detailed setting and methods are described in §5.3.

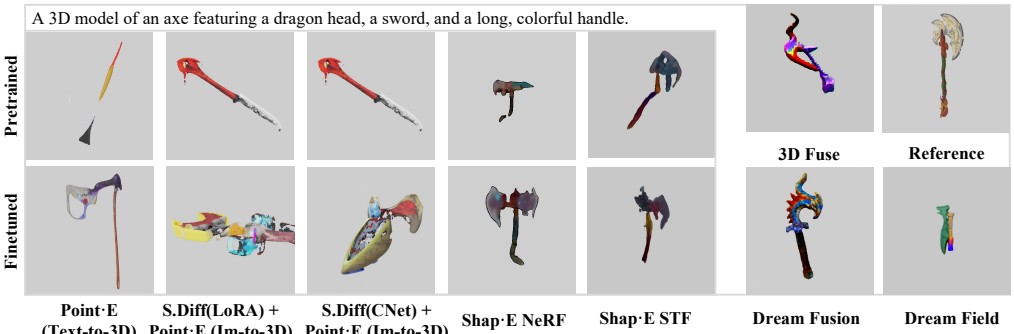

Figure 20: Text-to-3D results. The top text prompt and "Reference" are from our test set. We fine-tune the left 5-column methods on Cap3D-generated captions. The detailed setting and methods are described in §5.3.

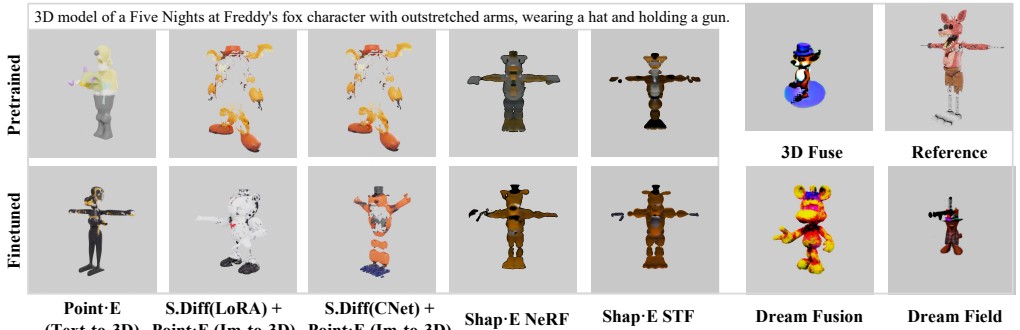

Figure 21: Text-to-3D results. The top text prompt and "Reference" are from our test set. We fine-tune the left 5-column methods on Cap3D-generated captions. The detailed setting and methods are described in §5.3.

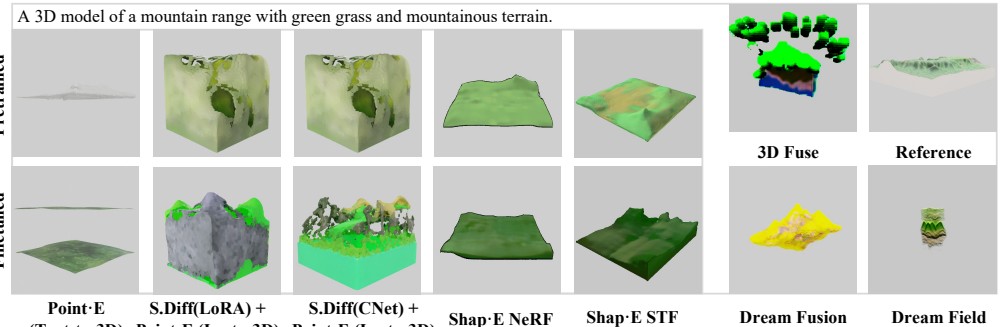

Figure 22: Text-to-3D results. The top text prompt and "Reference" are from our test set. We fine-tune the left 5-column methods on Cap3D-generated captions. The detailed setting and methods are described in §5.3.

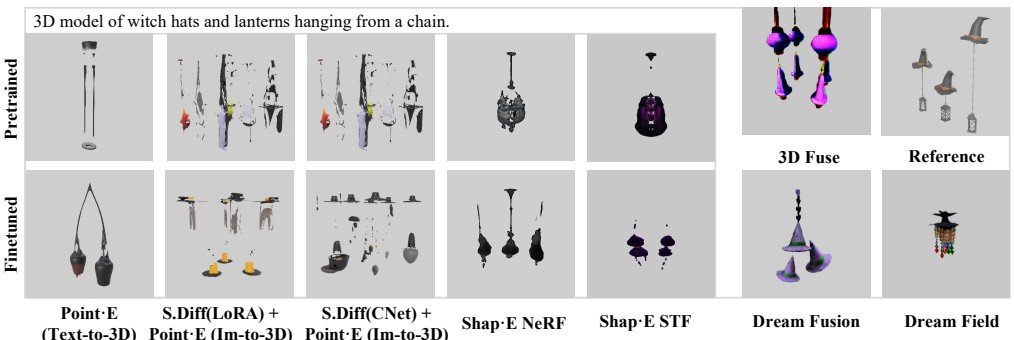

Figure 23: Text-to-3D results. The top text prompt and "Reference" are from our test set. We fine-tune the left 5-column methods on Cap3D-generated captions. The detailed setting and methods are described in §5.3.

## Appendix D    ABO Captioning: Automated Metrics

In §5.2, we report human A/B judgments on ABO. We do not report automated metrics, which are poor measures of performance for at least two reasons. First, ABO contains a large number of objects that are very similar, meaning it would be challenging for captions to distinguish their differences. Thus, retrieval metrics such as ViLT Image or Text Retrieval will show very poor scores across metrics. Second, we show automated captioning performs poorly at describing geometry well, meaning it is likely automated image-caption alignment will not align based on geometry well. For completeness, we report automated metrics in Table 8. As expected, all retrieval scores are very low. Automated captioning scores best across automated metrics, however we caution against drawing conclusions from this result. Human studies in Table 4 suggest the opposite, and qualitative results agree with this finding, e.g. Figure 4.

Table 8: **ABO Automated Caption Evaluations**. Automated captions are a poor measure of performance on ABO as (1) many objects are similar, making retrieval difficult; (2) automated captioning does not describe geometry well, so we should not expect automated image-caption alignment to describe geometrically correct captions well.

| Method | CLIP Score | ViLT Img Retr. R@5 | ViLT Img Retr. R@10 | ViLT Text Retr. R@5 | ViLT Text Retr. R@10 |
|---|---|---|---|---|---|
| Meta | 61.9 | 0.8 | 1.7 | 0.8 | 1.7 |
| Human | 75.2 | 2.6 | 4.4 | 2.3 | 4.2 |
| Cap3D | **89.9** | **4.2** | **7.2** | **3.2** | **5.6** |
| Cap3D(QA) | 82.7 | 2.9 | 5.3 | 2.4 | 4.3 |

In contrast with A/B tests, which take place on the full 6.4k objects of ABO, this table is computed on a random 5k object subset of ABO to follow standard retrieval benchmarks (performance drops considerably as dataset size increases. Using 5k instead of the full 6.4k makes it much easier to contextualize retrieval numbers). A/B performance on this 5k subset is very close to the full 6.4k dataset, meaning the sample is highly representative, and one can compare the results from this table in combination with Table 4 in the main paper.

## Appendix E    Cap3D, BLIP2, Human Caption Comparisons

A comparative analysis on the number of n-grams was conducted to shed light on the distinct vocabulary sizes among Cap3D, BLIP2, and human captions for Objaverse [9] objects. As elucidated in Table 9, Cap3D exhibits a considerably larger vocabulary as compared to BLIP2. Although human-generated captions encompass a higher number of phrases, the disparity is notably lesser than that observed between BLIP2 and Cap3D. Moreover, it is hypothesized that the slightly elevated dictionary size in human captions could be attributed to the inclusion of typographical errors, typically arising from crowdsourced platforms.

Table 9: N-gram Comparison among Cap3D, BLIP2, and Human Captions

| | Occ. in 5k captions | | |
|---|---|---|---|
| | Unigrams | Bigrams | Trigrams |
| BLIP2 | 1,928 | 6,616 | 10,899 |
| Cap3D | 3,108 | 15,274 | 25,883 |
| Human | 3,762 | 17,818 | 27,316 |

## Appendix F    Additional Details

### F.1    Prompt used in Cap3D

The two prompts used for BLIP2 used in Cap3D (QA) are (1) "Question: what object is in this image? Answer:" and (2) "Question: what is the structure and geometry of this <object>?" where <object> is replaced with the response to prompt (1).

For the prompt used in GPT4, we used "Given a set of descriptions about the same 3D object, distill these descriptions into one concise caption. The descriptions are as follows: 'captions'. Avoid describing background, surface, and posture. The caption should be:". We did several prompt engineering and considered prompt with more context, like "Below you will find a set of descriptions, each one is originating from various renderings of an identical 3D object. The level of accuracy in these descriptions ranges significantly: some might not correspond to the 3D object at all, others could be entirely accurate, while a few may only partially represent the object. Your task involves scrutinizing these descriptions and distilling them into a single, holistic depiction. The descriptions are as follows: 'captions'. Note: Please avoid using the phrases 'grey background', 'gray background', and 'gray surface' in your consolidated depiction. The synthesized description of the 3D object should be:". However, with those longer prompt with more context, we noticed GPT4 sometimes would generate its reasoning process which led to confusing output captions. Also, for the sake of cost, we hope to make our prompt as short as possible.

### F.2    Rendering Details

We use Blender to render 3D objects in Objaverse [9] and ABO [35]. For each object, we first normalize them into a unit cube and recenter to origin. Then, we place 8 different cameras surrounding the object with 2 cameras slightly below the object to capture the bottom of the object. Three area lights are placed and function as key light, fill light, and rim light, respectively. The detailed parameters are listed in our rendering script, provided in our Github.

In Objaverse, we filter out objects that fail in rendering, resulting a subset of 785k objects for rendering and captioning. In ABO, we exclude categories with simple geometry to concentrate on geometrical captioning, including "BLANKET", "RUG", "WALL_ART", "PLACEMAT", "CURTAIN", "MOUSE_PAD". This resulting a final subset of 6.4k objects for rendering and captioning.

### F.3    Human Captioning Split

Human captions are collected on a manually selected subset of Objaverse with good renders of nontrivial but decipherable objects. These objects are likely to be the most sensible for captioning and A/B testing. For instance, some Objaverse objects are essentially a simple rock with little texture; in others it can be difficult for a human to describe an object (e.g. abstract art, no clear object visible, or 3D scans with hard-to-distinguish details). These excluded objects are generally not effective samples to use for human A/B testing, as the correct caption may not be clear or may be trivial. We also exclude furniture, which is suitable for captioning, but we measure this with more focus on ABO. Human captions on ABO follow the split of [78].

## Appendix G    Crowdsourced Captioning Details

We use Hive for crowdsourced captioning. Workers are given instructions for the task including gold-standard examples. Captioning instructions are shared below for Objaverse in Figure 24 and ABO in Figure 25. Workers are persistently monitored. If a worker produces bad captions they are promptly banned from captioning, and their previous captions are discarded. Workers are paid approximately $50 per 1k tasks. We do not have access to their captioning rates; assuming a rate of 3 objects per minute, this would result in $9 per hour. Across Objaverse and ABO we spend a total of $7k on captioning.

Please describe the structure of the 3D object. Each answer should answer the following three questions.

(1) What pieces make up the object? (e.g. back, seat, legs, wheel, window)

(2) What shape are these pieces? (e.g. round, flat, square, long, narrow, wide, large)

(3) How are these pieces connected? (e.g. left, right, above, below, connected, held up by, on, inside of, has, contains)

**Please do not describe the color or texture (texture / color is purposely left out of renders).**

**Captions must be in English. If you do not provide real answers you will promptly be permanently banned; we actively monitor annotations**

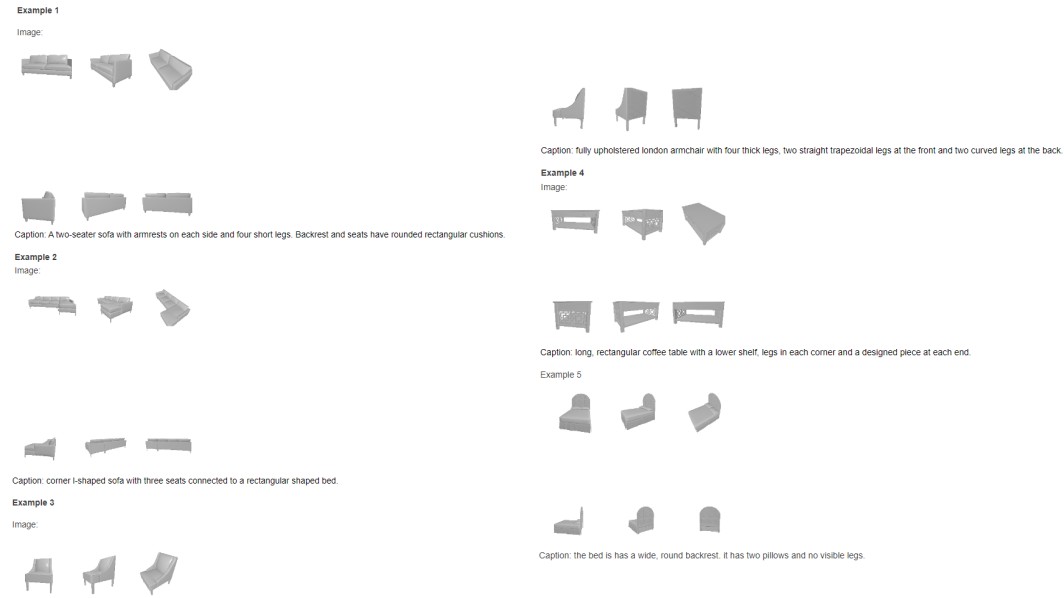

Example 1
Image:

Caption: A two-seater sofa with armrests on each side and four short legs. Backrest and seats have rounded rectangular cushions.

Example 2
Image:

Caption: corner l-shaped sofa with three seats connected to a rectangular shaped bed.

Example 3
Image:

Caption: fully upholstered london armchair with four thick legs, two straight trapezoidal legs at the front and two curved legs at the back.

Example 4
Image:

Caption: long, rectangular coffee table with a lower shelf, legs in each corner and a designed piece at each end.

Example 5

Caption: the bed is has a wide, round backrest. it has two pillows and no visible legs.

Figure 25: **ABO Caption Instructions.**

Please describe the object pictured in the image in terms of appearance and geometry in at least two sentences of detail.

**Captions must be in English. If you do not provide real answers you will promptly be permanently banned; we actively monitor annotations**

**Example 1**
Image:

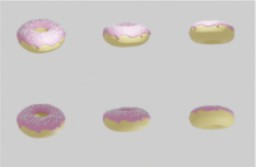

Caption: A cartoon donut that is vanilla with pink frosting and white sprinkles. The donut is round with a hole in the middle.
 --> Note: we describe in detail both appearance (cartoon donut, vanilla, pink frosting, white sprinkles) and geometry (round).

**Example 2**
Image:

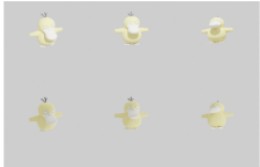

Caption: A yellow cartoon duck with a white beak and 3 dark hairs coming out of its head. It is the Pokemon Psyduck, and has a large round torso with short arms and legs.

--> Note: we describe in detail both appearance (yellow cartoon duck, white beak, dark hairs) and geometry (large round torso, short arms and legs, 3 hairs out of head). If you don't know specific names (e.g. Pokemon Psyduck) that is no problem! But if you do know, even better!

**Example 3**
Image:

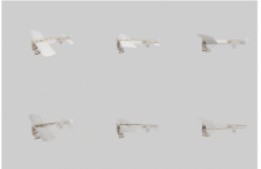

Caption: An early airplane model, it is very small and the frame is made of wood and the wings are made of a beige material. It has two wheels in the front and one in the back in the center.

--> Note: we describe in detail both appearance (airplane model, frame is wood, wings beige) and geometry (very small, two wheels in front, one wheel in back in center).

**Example 4**
Image:

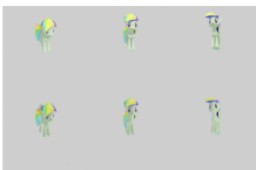

Caption: A light-green toy pony with colorful hair on its head and tail. It has big eyes and stars on its cheeks and fluffy hair and tail.

--> Note: we describe in detail both appearance (light-green toy pony, colorful hair on head and tail, stars on cheeks) and geometry (big eyes, fluffy hair and tail).

Figure 24: **Objaverse Caption Instructions.**

## Appendix H    Crowdsourced A/B Testing Details

We use Hive for crowdsourced A/B testing. Specifically, workers are given an image and two captions, and select which is better on a scale from 1 to 5, where 3 is a tie. So 1 would be "left much better", and 2 would be "left better". Workers are given instructions for the task along with gold standard examples. Workers are informed to prioritize accuracy, then informative detail, then brevity. Left/right order between methods was randomized for each instance. A/B Testing instructions are shared below for Objaverse in Figure 27 and ABO in Figure 26.

Workers are automatically banned by the platform if they miss too many gold-standard examples. However, we found some workers would successfully pass the handful of gold-standard examples while scamming on the rest of the examples. The most common scam cases were always picking the same number, or always picking the shorter or longer caption. We thus manually search through all workers and ban workers who meet these scamming criteria and discard their judgments. Unfortunately, discarding judgments leads to uneven numbers of observations for each individual experiment. Nevertheless, in all cases, enough observations are available to draw conclusive findings.

The size of each experiment's data after discarded judgments is below.

- *Objaverse Split (1)* takes place on a random set upon which human captions are available. *Cap3D vs. Human* has 36k observations across 22k objects.

- *Objaverse Split (2)* takes place on a random object set upon which human captions are available. *Cap3D vs. Human* has 10k observations across 4.7k objects. *Cap3D vs. Metadata* has 7k observations across 4.7k objects (less than the target 10k), though given the extremely poor rating of Metadata, results are conclusive.

- *Objaverse Split (3)* takes place on a random object set upon the entire Objaverse dataset. *Cap3D vs. BLIP2* has 20k observations across 5.0k objects and *Cap3D vs. +GPT4* has 29k observations across 5.0k objects.

- *ABO* takes place on the full ABO object set. *Human vs. Cap3D* has 21k observations across 6.4k objects, *Cap3D (QA) vs. Human* has 17k observations across 6.4k objects, *Cap3D (QA) vs. Cap3D* has 13k observations across 6.4k objects, and *Cap3D (QA) vs. Meta* has 12k observations across 6.4k objects.

Workers are paid approximately $20 per 1k tasks. We do not have access to their captioning rates; assuming a rate of 7.5 A/B tests selected per minute, this would result in $9 per hour. Across Objaverse and ABO we spent a total of $1.8k on A/B testing.

In this task, an object will be pictured from 8 views. There will be two captions ("left" caption and "right" caption).

Please select the caption best describing the pictured object in terms of **type, appearance, and structure.**

- Prioritize captions that make **accurate statements**.
- Select captions with **clear, informative object details**
- **Ignore** caption details about the background (these are neither good nor bad to the task)
- NOTE: There is only one object. It appears 8 times since it is shown from 8 views
**----> If a caption says there are multiple objects and there is only one object in each of 8 views, the caption is wrong**

Select from the following options:

1. Left Much Better - left caption describes the object with much more accuracy, informative detail, or avoiding many unnecessary details

2. Left Better: left caption describes the object with more accuracy, informative detail, or avoiding unnecessary details
3. Tie: both captions have similar accuracy and detail

4. Right Better: right caption describes the object with more accuracy, informative detail, or avoiding unnecessary details

5. Right Much Better: right caption describes the object with much more accuracy, informative detail, or avoiding many unnecessary details

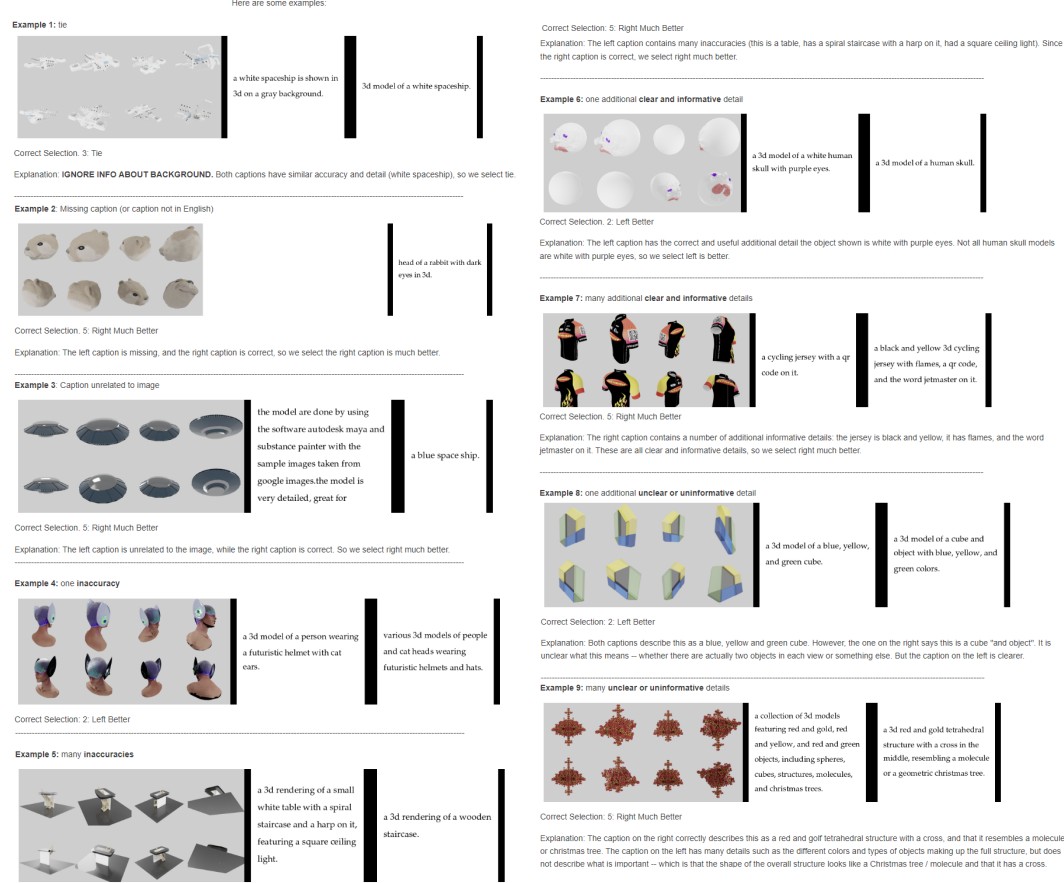

Figure 26: **A/B Instructions: Objaverse Captions.**

In this task, an object will be pictured from 6 views. There will be two captions ("left" caption and "right" caption).

Please select the caption best describing the **structure of the pictured object**

- **Language describing the color or texture of the object or the background should be ignored -- only judge based on object geometry.**
- Prioritize captions that make **accurate statements about the object**.
- Select captions with **informative object details** but **not ones with unrelated or repeated (redundant) details**.
- NOTE: There is only one object. It appears 6 times since it is shown from 6 views
- ----> **If a caption says there are multiple objects and there is only one object in each of 6 views, the caption is wrong**

Select from the following options:

1. Left Much Better: left caption describes the object geometry with much more accuracy, informative detail, or avoiding many unnecessary details

2. Left Better: left caption describes the object geometry with more accuracy, informative detail, or avoiding unnecessary details
3. Tie: both captions have similar accuracy and detail

4. Right Better: right caption describes the object geometry with more accuracy, informative detail, or avoiding unnecessary details

5. Right Much Better: right caption describes the object geometry with much more accuracy, informative detail, or avoiding many unnecessary details

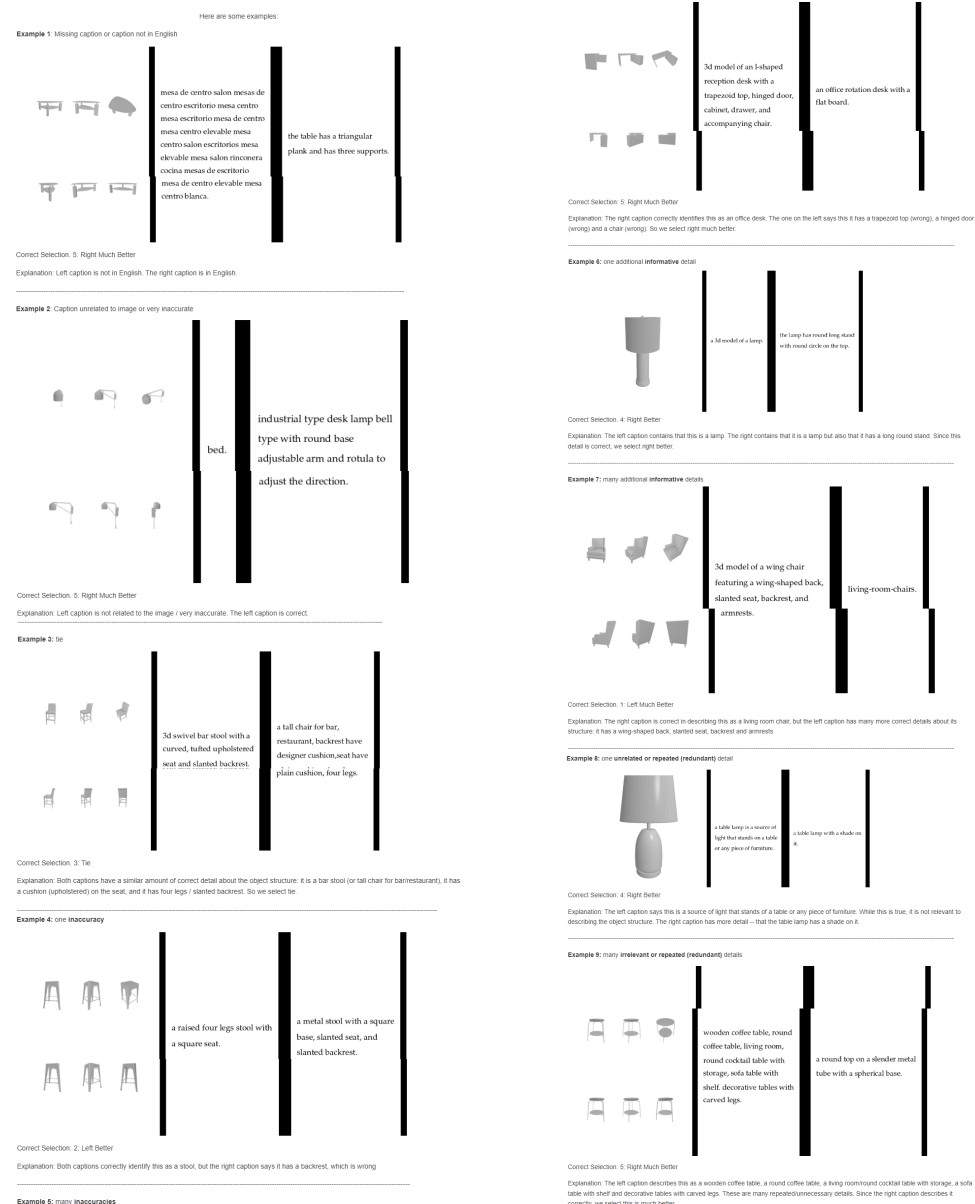

Figure 27: **A/B Instructions: ABO Captions.**

# Appendix I   Additional Experimental Details

**Captioning**: we perform one full-scale evaluation run for all captioning experiments; 95% confidence interval for mean is presented. Metrics are overviewed in §5.1; A/B testing is detailed further in §H. CLIP Score takes about 5 minutes, while ViLT R-Precision takes about 8 hours using an A40 for test set of 5k object-caption pairs. Crowdsourced A/B testing takes about 12 hours for 10k responses across 5k objects.

**Text-to-3D, finetuning**: for finetuning experiments, we used one train and evaluation run using a learning rate validated on a small overfitting experiment on the train set. Training took about 3 days on the full set and 1 day on the small (human) set. We used AdamW optimizer and CosineAnnealingLR scheduler with initial learning rate $1e-5$ for finetuning both Point·E and Shap·E. We adopted batch size $64$ and $256$ for Shap·E and Point·E, respectively. However, for Shap·E, we found it usually outputs NaN and needed to re-start from saved checkpoints, which could be one of the reaons why our finetune did not bring improvements. For LoRA, we use AdamW optimizer and CosineAnnealingLR scheduler with initial learning rate $1e-4$ and batch size of 3. For ControlNet, we use AdamW optimizer and constant learning rate of $1e-5$ and batch size of 8. Experiments use 4 A40s to train except LoRA, which fails upon multi-gpu training due to a HuggingFace internal DDP error. Notably single-gpu training still yields improvement. Evaluation takes the following time (in seconds) per iteration, which includes rendering:

- PointE (text-to-3D): 37sec = 28sec (text-to-3D) + 9sec (render)
- LoRA + PointE(im-to-3D): 114sec = 5sec + 100sec (im-to-3D) + 9sec (render)
- ControlNet + PointE(im-to-3D): 124sec = 15sec + 100sec (im-to-3D) + 9sec (render)
- ShapE (NeRF): 193sec (text-to-3D + render)
- ShapE (stf): 16sec (text-to-3D + render)

Note publicly available PointE (im-to-3D) is 1B param, making it slower than the largest publicly available PointE (text-to-3D) of 40M. Evaluation metrics are detailed in §5.3.

**Text-to-3D, optimization**: For one object, optimization plus final rendering takes 40 minutes for 3DFuse, 95 minutes for Stable DreamFusion, and 35 minutes for DreamField; using 1 A40 GPU. We use default parameters for all methods and run them once.

