# OpenReview forum: "Scalable 3D Captioning with Pretrained Models"
_NeurIPS.cc/2023/Track/Datasets_and_Benchmarks — NeurIPS 2023 Datasets and Benchmarks Poster_

### Official Review · Reviewer_Dwxs · 2023-07-12
**The paper is well-written. The method of dataset generation is effective. While the technical contributions are a little weak.**

**Rating:** 7
**Confidence:** 4
**Correctness:** Yes. The dataset is constructed in a …
**Clarity:** Yes

**Strengths:**

The submission proposes to utilize GPT-4 to fuse captions generated from multi-view rendered images, which is a novel idea. Besides, the submission does much ethical filtering work. Experiments on Objaverse and ABO dataset demonstrate advantages over metadata and human-authored captions, and effectiveness in finetuning text-to-3D models on the generated dataset.

**Additional Feedback:**

No

**Documentation:**

Yes.

**Limitations:**

The technical contribution is a littl weak.The idea of generating text-3D paired data using LLMs is not novel. Compared with Ref.[86], the submission additionly conducts the fusion precedure using GPT-4, does much ethical filtering work, and conducts text-to-3D tasks on the generated dataset.

**Opportunities For Improvement:**

The proposed method only considers Objaverse and ABO dataset. Is it possible to utilize the proposed method on other available 3D datasets, to bulid a larger text-3D paired dataset？

**Relation To Prior Work:**

Yes. Within the related work, the authors illustrated that Ref.[86] also conducted 3D obejct caption, while they do not fuse captions across views and do not approach text-to-3D.

**Summary And Contributions:**

This paper introduces a method to generate descriptive text for 3D objects using LLMs, and contribute a dataset of 660k 3D-text pairs. Evaluations and experiments demonstrate advantages over human-annotations, and effectiveness in text-to-3d tasks.

---

> ### Author Response · Authors · 2023-08-16
> **Author Response**
>
> We thank Reviewer Dwxs for the valuable reviews and insightful comments. We address your concerns below and will incorporate our final discussion into the revision.
>
> ---
> Q1:
> > The proposed method only considers Objaverse and ABO dataset. Is it possible to utilize the proposed method on other available 3D datasets, to bulid a larger text-3D paired dataset？
>
> A: Definitely! As we plot in the introduction, Cap3D is agnostic to 3D asset sources and can be effectively scaled to larger extents with increased 3D assets and computational resources. For example, we can apply Cap3D on Objaverse-XL. Furthermore, our method can help filter out 3D objects in Objaverse-XL with ethical issues as described in Section 3.2.
>
>
> ---
> Q2:
> > The technical contribution is a littl weak.The idea of generating text-3D paired data using LLMs is not novel. Compared with Ref.[86], the submission additionly conducts the fusion precedure using GPT-4, does much ethical filtering work, and conducts text-to-3D tasks on the generated dataset.
>
> A: We emphasize that [86] is unpublished and first appeared on arXiv on 5/14/2023, less than one month before the NeurIPS Datasets & Benchmarks deadline on 6/7/2023. [Per the NeurIPS 2023 FAQ](https://neurips.cc/Conferences/2023/PaperInformation/NeurIPS-FAQ), [86] should be viewed as concurrent with our work, and it is thus inappropriate to penalize our work for proposing similar ideas as [86].
>
> Even so, compared to [86] our paper:
> - Uses GPT-4 to fuse captions from multiple views, vs using a set of per-view captions;
> - Implementing ethical filters on 3D assets using rendered images and generated captions;
> - Addresses the task of text-to-3D generation, vs 3D classification;
> - Provides careful comparisons and user studies to measure the difference between our generated captions, those written by human crowd workers, and those available in 3D asset metadata (Figures 3, 4, Tables 3, 4, 5).

---

> ### Author Response · Authors · 2023-08-28
> **Gentle Reminder**
>
> Dear Reviewer Dwxs,
>
> We appreciate the time and effort you’ve already invested in reviewing our work. As the rebuttal review deadline is approaching (***1 day left***), we wanted to send a gentle reminder to review our rebuttal. We believe we have addressed all the points raised in the initial review and hope that our explanations will clarify any remaining questions or concerns. Please do not hesitate to reach out if you need any additional information or have further questions.
>
> Thank you once again for your kind consideration.

---

### Official Review · Reviewer_UuMn · 2023-07-21
**Review to Scalable 3D Captioning**

**Rating:** 7
**Confidence:** 4

**Strengths:**

1.	This work addresses an important problem by focusing on generating descriptive text for 3D objects, effectively bridging the gap between 3D objects and text modalities.
2.	By conducting fine-tuning on several text-to-3D models, the dataset facilitates meaningful improvements in the performance of these models. This dataset plays a critical role in evaluating and advancing text-to-3D generation.
3.	The evaluation results presented in the paper showcase the significant performance of the proposed method when compared to tedious and low-quality human annotations.

**Additional Feedback:**

Thanks for the reply.  The authors have addressed most of my concerns and I would like to keep my rating.

**Clarity:**

This paper is well written. This paper presents a clear and concise description of the problem.

**Correctness:**

I believe that the claims and dataset construction appear to be well-founded and reliable.

**Documentation:**

There is sufficient detail on data collection and organization, availability and maintenance, and ethical and responsible use. This dataset is entirely based on existing public datasets and follows the standards of these datasets.

**Ethics:**

I do not see any obvious ethical concerns about this paper.

**Limitations:**

1 This work only uses rendered images information from 3D objects as input, which may result in the loss of some 3D information, such as geometry details obscured by occlusion, part functions, and accurate position information.

2 The pipeline designed in this work is not end-to-end, but rather a combination of multiple existing solutions. While this approach might achieve the desired goal, an end-to-end solution would potentially streamline the process, reducing complexity and improving efficiency.

3 The comparison solely based on human-authored captions and their associated classes might have limitations. Human descriptions can be subjective and might not comprehensively capture all properties of the 3D objects. Including a qualitative comparison that evaluates various aspects like categories, colors, textures, shapes, components, and functions could provide a more comprehensive assessment of the model's performance

**Opportunities For Improvement:**

This work utilizes blip, clip, and gpt4 to generate text captioning results. However, given the inherent diversity of these models, the generation results should be various each time. It is crucial for the authors to verify the differences and assess the robustness of the various generation results.

**Relation To Prior Work:**

The introduction and related work sections nicely discussed relevant work and datasets. The contribution of this work is apparent.

**Summary And Contributions:**

The authors propose Cap3D, an automatic framework for generating descriptive captions of 3D objects using pretrained models. The method involves several steps: firstly, rendering eight 2D images for each 3D object; then, utilizing BLIP to generate five paragraphs of descriptive captioning for each image; further employing CLIP for precise text selection; and finally, leveraging GPT4 to aggregate the captions from various perspectives.
To validate the quality of Cap3D, evaluations were conducted on the Objaverse and ABO datasets. The results show that Cap3D can produce significant fine-grained geometric detail and generally longer captions when compared to human-authored annotations.

Overall, this work offers valuable insights into the generation of descriptive text for 3D objects, with promising outcomes from the proposed dataset and evaluation results.

---

> ### Author Response · Authors · 2023-08-16
> **Author Response (1/3)**
>
> We thank Reviewer UuMn for the valuable reviews and insightful suggestions. We address your concerns below and will incorporate our final discussion into the revision.
>
> ---
> Q1:
> > This work utilizes blip, clip, and gpt4 to generate text captioning results. However, given the inherent diversity of these models, the generation results should be various each time. It is crucial for the authors to verify the differences and assess the robustness of the various generation results.
>
> A:
> Thank you for the insightful comments. During rebuttal, we studied the robustness of the Cap3D generation via both quantitative and qualitative results. We find our outputs are generally the same or share high semantic similarity to each other.
>
> ### Quantitative Analysis:
> We use Cap3D on each of the 200 objects ten times, generating a total of 2,000 captions. We then evaluate the results using three metrics: (1) caption uniqueness:  quantifies how many generated captions for the same object are exactly the same (***smaller means more similar, upperbound 10***); (2) semantic similarity: measure through Sentence-BERT embedding distance (***smaller means more similar***); (3) semantic similarity: measure through cosine similarity based on Term Frequency-Inverse Document Frequency (TF-IDF,  ***bigger means more similar, upperbound 1***).
>
> The results over the randomly sampled 200 objects are in the below table. To better understand the statistics, we randomly visualize several qualitative results below.
>
> | Uniqueness | Semantic similarity (TF-IDF) | Semantic similarity (Sentence-BERT) |
> |---------------------|----------|-------------------|
> | 5.77                  | 0.766  | 0.324   |
>
>
>
> ### Sampled Qualitative Results:
>
> ---
>
> ***Uid***: 822202b6b8594342b632b0a39432642a
>
> ***Image***: https://ltg-webaccess-files.s3.us-east-2.amazonaws.com/exp_r3q11.png
>
> [
> - "3D model of a black and white pixel bike"
> - "3D model of a pixelated, black and white bicycle."
> - "3D model of a pixelated, black and white bicycle."
> - "3D model of a pixelated, black and white bicycle"
> - "3D model of a pixelated, black and white bicycle."
> - "3D model of a pixelated, black and white bicycle."
> - "3D model of a pixelated, black and white bicycle."
> - "3D model of a pixelated, black and white bicycle."
> - "3D model of a pixelated, black and white bicycle."
> - "3D model of a pixelated, black and white bicycle"
> ]
>
> | Uniqueness | Semantic similarity (TF-IDF) | Semantic similarity (Sentence-BERT) |
> |---------------------|----------|-------------------|
> | 3 (the last caption lacks a .)           |  0.811  	| 0.38   |
>
> ---
>
> ***Uid***: 2968ed36911043359cc110067ab8b725
>
> ***Image***: https://ltg-webaccess-files.s3.us-east-2.amazonaws.com/exp_r3q12.png
>
> [
> - "3D model of an old building with a door, stairs, and a window."
> - "3D model of an old building with a door, stairs, and a window."
> - "3D model of an old building with a door, stairs, and a window."
> - "3D model of an old building with a door, stairs, and a window."
> - "3D model of an old building with a door, stairs, and a window."
> - "3D model of an old building with a door, stairs, and a window."
> - "3D model of an old building with a door, stairs, and a window."
> - "3D model of an old building with a door, stairs, and a window."
> - "3D model of an old building with a door, stairs, and a window."
> - "3D model of an old building with a door, stairs, and a window."
> ]
>
> | Uniqueness | Semantic similarity (TF-IDF) | Semantic similarity (Sentence-BERT) |
> |---------------------|----------|-------------------|
> | 1                  | -  | -   |
>
> ---
>
> ***Uid***: b416732d162e4e4d92ee9a37c65fedbc
>
> ***Image***: https://ltg-webaccess-files.s3.us-east-2.amazonaws.com/exp_r3q13.png
>
> [
> - "3D model of a black desk lamp with a blue handle and light."
> - "3D model of a black desk lamp with a blue handle and blue light"
> - "3D model of a black desk lamp with a blue handle and blue light."
> - "3D model of a black desk lamp with a blue handle and blue light."
> - "3D model of a black desk lamp with a blue handle and blue light."
> - "3D model of a black desk lamp with a blue handle and blue light."
> - "3D model of a black desk lamp with a blue handle and blue light."
> - "3D model of a black desk lamp with a blue handle and blue light."
> - "3D model of a black desk lamp with a blue handle and blue light."
> - "3D model of a black desk lamp with a blue handle and blue light."
> ]
>
> | Uniqueness | Semantic similarity (TF-IDF) | Semantic similarity (Sentence-BERT) |
> |---------------------|----------|-------------------|
> | 3                  | 0.985  | 0.126   |

---

> > ### Author Response · Authors · 2023-08-16
> > **Author Response (2/3)**
> >
> > ***Uid***: 3b6197d570974a07ac9cc4e8b5feb1af
> >
> > ***Image***: https://ltg-webaccess-files.s3.us-east-2.amazonaws.com/exp_r3q14.png
> >
> > [
> > - "3D model of a farm with green fields and trees"
> > - "3D model of a farm with green fields, trees, and grassy areas."
> > - "3D model of a farm with green fields, trees, and grassy areas."
> > - "3D model of a farm with green fields, trees, and grassy areas."
> > - "3D model of a green, grassy farmland with fields and trees."
> > - "3D model of a green, grassy farmland with fields and trees."
> > - "3D model of a green, grassy farmland with fields and trees."
> > - "3D model of a green, grassy farmland with fields and trees."
> > - "3D model of a green, grassy farmland with fields and trees."
> > - "3D model of a green, grassy farmland with fields and trees."
> > ]
> >
> > | Uniqueness | Semantic similarity (TF-IDF) | Semantic similarity (Sentence-BERT) |
> > |---------------------|----------|-------------------|
> > | 3                  | 0.839  | 0.47   |
> >
> > ---
> >
> > ***Uid***: 862b7c8cae7e4f8ebc98ee3fd71bdc39
> >
> > ***Image***: https://ltg-webaccess-files.s3.us-east-2.amazonaws.com/exp_r3q15.png
> >
> > [
> > - "3D rendering of a black wall with a window, featuring a yellow building and box."
> > - "3D rendering of a black wall with a window, featuring a yellow building and a box."
> > - "3D rendering of a black wall with a window, featuring a yellow building and a box."
> > - "3D rendering of a black wall with a window, featuring a yellow building and various boxes."
> > - "3D rendering of a black wall with a window, featuring a yellow box and a small yellow building."
> > - "3D rendering of a black wall with a window, featuring a yellow building and various boxes."
> > - "3D rendering of a black wall with a window and a yellow building, featuring various boxes."
> > - "3D rendering of a black wall with a window, featuring a yellow building and various boxes."
> > ]
> >
> > | Uniqueness | Semantic similarity (TF-IDF) | Semantic similarity (Sentence-BERT) |
> > |---------------------|----------|-------------------|
> > | 7                  | 0.692  | 0.329   |
> >
> > ---
> >
> > ***Uid***: 0290737f64b14b08b87faa644580f3a9
> >
> > ***Image***: https://ltg-webaccess-files.s3.us-east-2.amazonaws.com/exp_r3q16.png
> >
> > [
> > - "3D model of a four-legged shelving unit or storage rack"
> > - "3D model and rendering of a shelving unit with a suspended ceiling system and hanging shelf."
> > - "3D model and rendering of a shelving unit with a suspended ceiling system and hanging shelf."
> > - "3D model and rendering of a shelving unit with storage rack and suspended ceiling system."
> > - "3D model and rendering of a shelving unit with storage rack, suspended ceiling system, and hanging shelf features."
> > - "3D model and rendering of a shelving unit with storage rack, suspended ceiling system, and hanging shelf."
> > - "3D model and rendering of a shelving unit with storage rack, suspended ceiling system, and hanging shelf features."
> > - "3D model and rendering of a shelving unit with storage rack, suspended ceiling system, and hanging shelf features."
> > - "3D model and rendering of a shelving unit with storage rack, suspended ceiling system, and hanging shelf features."
> > - "3D model and rendering of a shelving unit with storage rack, suspended ceiling system, and hanging shelf features."
> > ]
> >
> > | Uniqueness | Semantic similarity (TF-IDF) | Semantic similarity (Sentence-BERT) |
> > |---------------------|----------|-------------------|
> > | 5                  | 0.728  | 0.405   |

---

> > > ### Author Response · Authors · 2023-08-16
> > > **Author Response (3/3)**
> > >
> > > ---
> > > Q2:
> > > > This work only uses rendered images information from 3D objects as input, which may result in the loss of some 3D information, such as geometry details obscured by occlusion, part functions, and accurate position information.
> > >
> > > A: While it is recognized that using rendered images from 3D objects as input may result in the loss of some 3D details, this is also the way humans perceive 3D objects, without always having access to tactile information. Furthermore, given the lack of models trained on massive 3D-text data, rendering offers a viable alternative by enabling the use of existing image-text models trained at scale. Although it is out of the scope of our paper, it would be interesting to train 3D-text captioning using our large-scale generated dataset. We hope our dataset could help future work approach this task.
> > >
> > > ---
> > > Q3:
> > > > The pipeline designed in this work is not end-to-end, but rather a combination of multiple existing solutions. While this approach might achieve the desired goal, an end-to-end solution would potentially streamline the process, reducing complexity and improving efficiency.
> > >
> > > A: While the aspiration towards an end-to-end solution is acknowledged, our current approach strategically combines existing solutions to address the inherent chicken-and-egg problem. As we progress, Cap3D may pave the way for an end-to-end framework, given the prerequisite of paired 3D-text data.
> > >
> > > ---
> > >
> > > Q4:
> > > > The comparison solely based on human-authored captions and their associated classes might have limitations. Human descriptions can be subjective and might not comprehensively capture all properties of the 3D objects. Including a qualitative comparison that evaluates various aspects like categories, colors, textures, shapes, components, and functions could provide a more comprehensive assessment of the model's performance
> > >
> > > A: Thanks for the suggestion! We agree breaking down comparison into components is helpful. We compared 500 examples of human-authored vs. Cap3D-generated captions, rating them 1-5 for the categories below:
> > >
> > > | Categories  | Human Score (1-5) vs. Cap3D | Human Win % vs. Cap3D | Human Lose % vs. Cap3D |
> > > |-------------|-----------------------------|-----------------------|------------------------|
> > > | Colors      | 2.95 ± 0.04                 | 24.6% ± 1.9%          | 28.2% ± 2.0%           |
> > > | Textures    | 2.70 ± 0.03                 | 4.6% ± 0.9%           | 34.0% ± 2.1%           |
> > > | Shapes      | 2.57 ± 0.03                 | 9.0% ± 1.3%           | 48.4% ± 2.2%           |
> > > | Components  | 2.64 ± 0.04                 | 21.8% ± 1.8%          | 55.6% ± 2.2%           |
> > > | Functions   | 2.994 ± 0.004               | 0.2% ± 0.2%           | 0.8% ± 0.4%            |
> > >
> > >
> > > Human annotators tend to do a relatively good job at colors and a worse job at other classes, compared to Cap3D. This is due to the relative ease of a human naming colors of unknown objects, while misunderstanding the specific object, and thus giving less description to other features such as category, shape and components. For instance, in the below example, the human caption gets the color correct but does not know the precise name of the structure (cabinet) and instead uses the approximate description “a cubical brown box with four… legs”. In contrast, Cap3D correctly describes this as a cabinet, adding it has two doors – which improves over human in category (cabinet) and components (cabinet with two doors).
> > >
> > > ***Uid***: 4efbff8671d24f689dafb03e549b4135
> > >
> > > ***Image***: https://ltg-webaccess-files.s3.us-east-2.amazonaws.com/exp_r3q41.png
> > >
> > > ***Human Caption***: a cubical brown box with four small standing legs.
> > >
> > > ***Cap3D Caption***: a 3D model of a brown cabinet with two doors, doubling as a TV stand and bench.
> > >
> > > Consider the following example. While the human caption is correct, Cap3D more precisely identifies this as a specific model of car, which gives more information on category (Delorean), and implicitly informing texture (texture of a Delorean: stainless steel) shape (the shape of a Delorean: sleek, square design), and components (flux capacitor from Back to the Future Movie).
> > >
> > > ***Uid***: 18ecd0636aaa4373a903b76f9050c291
> > >
> > > ***Image***: https://ltg-webaccess-files.s3.us-east-2.amazonaws.com/exp_r3q42.png
> > >
> > > ***Human Caption***: Gray coloured toy car with black tires.
> > >
> > > ***Cap3D Caption***: “Back to the Future Delorean Car 3D model.”
> > >
> > > We note the goal of our captioning isn’t functionality, and we don’t put this in the instructions for Cap3D or human annotations. For completeness, we test this and find Human and Cap3D usually ties, as typically neither describes functionality.

---

> ### Author Response · Authors · 2023-08-28
> **Gentle Reminder**
>
> Dear Reviewer UuMn,
>
> We appreciate the time and effort you’ve already invested in reviewing our work. As the rebuttal review deadline is approaching (***1 day left***), we wanted to send a gentle reminder to review our rebuttal. We believe we have addressed all the points raised in the initial review and hope that our explanations will clarify any remaining questions or concerns. Please do not hesitate to reach out if you need any additional information or have further questions.
>
> Thank you once again for your kind consideration.

---

### Official Review · Reviewer_bhAA · 2023-07-30
**Good paper but with some limitations**

**Rating:** 6
**Confidence:** 5
**Correctness:** Yes
**Clarity:** Yes

**Strengths:**

1. A large-scale 3D text-object paired dataset is in short supply and important for 3D reasoning study. The authors generate such a dataset and publicly release it.

2. The proposed automatic annotation method is reasonable and straightforward. The automatically generated caption surpasses human-authored descriptions in terms of quality, cost, and speed.

**Additional Feedback:**

If the caption contains more 3D-specific information (e.g., spatial relationship), it may be benefit for more downstream 3D reasoning tasks.

**Documentation:**

Yes

**Limitations:**

Yes,  the authors have adequately addressed the limitations and potential negative societal impact of their work

**Opportunities For Improvement:**

1. The generated caption is too simple. Most captions in objaverse dataset only contain category information with simple attributes. However, this information may be more suitable for 2D image caption instead of 3D. The spatial relationship in the 3D object is not considered.

2. Some images in specific viewpoints may generate incorrect information and the CLIP may not detect it. For example, when we render an image of a IPAD exactly from the side view, the image only contains a rectangle. BLIP2 may generate incorrect information. In addition, the CLIP may consider this incorrect information aligned with the image. When summarizing the information using GPT-4, there is no specific design for filtering incorrect information. The GPT-4 may misunderstand there are two different objects in the 3D scene and generate a wrong caption.

**Relation To Prior Work:**

Yes

**Summary And Contributions:**

This paper proposes to use the ability of large foundation models for captioning a large-scale 3D object dataset. Specifically, the authors first render several 2D images of the 3D objects from different viewpoints. For each image, a BLIP2 is leveraged to generate multiple captions, and a CLIP is leveraged to filter out the misaligned caption. Last, a GPT-4 is leveraged for summarizing these multi-view captions to get the final 3D object caption. The automatically generated caption surpasses human-authored descriptions in terms of quality, cost, and speed. The proposed method is reasonable and straightforward. However, I am concerned about some limitations listed below.

---
### After Rebuttal
We thank the authors for the detailed response which solves most of my concerns. Using GPT-4 with a carefully designed prompt for summarizing is helpful for filtering out incorrect information, but designing some explicit methods may perform better.  Nevertheless, this paper is a good paper and the released data is of good quality and helpful for the research community. Thus, I lean to accept this paper for publication.

---

> ### Author Response · Authors · 2023-08-16
> **Author Response**
>
> We thank Reviewer bhAA for the valuable reviews and comments. We address your concerns below and look forward to your replies. We will incorporate our final discussion into the revision.
>
> ---
> Q1:
> > The generated caption is too simple. Most captions in objaverse dataset only contain category information with simple attributes. However, this information may be more suitable for 2D image caption instead of 3D. The spatial relationship in the 3D object is not considered.
>
> A: We appreciate the feedback on the simplicity of our generated captions. However, a closer look at our results reveals a different narrative. Figure 3 showcases that Cap3D captions often possess more detail than human-generated ones. Our captions do not solely rely on 2D information but consolidate information from multiple views, as evident from Figure 1, where [row1 col3] "skull" and "pizza" are depicted, aggregating insights from various angles and [row3 col1] “purple spikes on the back” are not shown in the first picture. More examples are included in Appendix B.
>
> Furthermore, while our primary focus is on single 3D objects, it's essential to note that we have incorporated geometric captioning (***Section 5.2***), which might be more relevant for our dataset than emphasizing spatial relationships (often considered in broader 3D scenes). By incorporating BLIP-2 QA, we have managed to produce captions with similar-to-more detail than human captions, including detailed geometrical information of the 3D object, such as "sofa with backrest and armrests" (Figure 4). Other examples can be found in Figure 1 (row 4 col 3 & 4), with the description "sofa... with a footstool on one side."
>
> ---
> Q2:
> > Some images in specific viewpoints may generate incorrect information and the CLIP may not detect it. For example, when we render an image of a IPAD exactly from the side view, the image only contains a rectangle. BLIP2 may generate incorrect information. In addition, the CLIP may consider this incorrect information aligned with the image. When summarizing the information using GPT-4, there is no specific design for filtering incorrect information. The GPT-4 may misunderstand there are two different objects in the 3D scene and generate a wrong caption.
>
> A: We share this concern, which is why we employ eight viewpoints for each object caption, covering both top and bottom as well as horizontal information. This approach filters out rare, extreme viewpoints by utilizing a voting system to eliminate 'bad' captions.
>
> Additionally, we find two real cases in our dataset which are very close to what you described. Both of them validate our use of multiple view images for robust to “bad” viewpoints.
>
>
> ### Case1:
>
> ***Uid***: 9b4f65ed3ba54bce84b5508392c3e8aa
>
> ***Image***: https://ltg-webaccess-files.s3.us-east-2.amazonaws.com/exp_r2q21.png
>
> For the bottom-right rendered images, BLIP2 generated
>
> [
> - 'a 3d model of a laptop on a gray background',
> - 'a 3d model of a laptop',
> - 'ipad air 2 - 3d model - preview no 1',
> -  'a 3d model of a laptop on a gray background',
> -  'a 3d model of a laptop on a gray background',
> ]
>
> and CLIP selected the first one [ 'a 3d model of a laptop on a gray background'], which is inaccurate. However, after voting via GPT4, the final caption for this object is ***iPad Air 2***, as most of the captions from other views are accurate.
>
>
>
>
> ### Case2:
>
> ***Uid***: a8024eb337e24596bb18231ffe511e4d
>
> ***Image***: https://ltg-webaccess-files.s3.us-east-2.amazonaws.com/exp_r2q22.png
>
> This case shares similar insights as case 1. For the bottom-right rendered images, BLIP2 generated
>
> [
> - 'a 3d rendering of a large gray box',
> - 'a 3d rendering of a flat surface',
> -  'a 3d rendered image of a white square on a gray background',
> - 'a 3d rendering of a white square on a gray background',
> -  'a 3d rendering of a white square on a gray background',
> ]
>
> and CLIP selected the 4th caption [''a 3d rendering of a white square on a gray background''], which is inaccurate. However, after voting via GPT4, the final caption for this object is ***a white iPad Air 2 tablet***.

---

> ### Author Response · Authors · 2023-08-28
> **Gentle Reminder**
>
> Dear Reviewer bhAA,
>
> We appreciate the time and effort you’ve already invested in reviewing our work. As the rebuttal review deadline is approaching (***1 day left***), we wanted to send a gentle reminder to review our rebuttal. We believe we have addressed all the points raised in the initial review and hope that our explanations will clarify any remaining questions or concerns. Please do not hesitate to reach out if you need any additional information or have further questions.
>
> Thank you once again for your kind consideration.

---

### Official Review · Reviewer_WJz6 · 2023-07-31

**Rating:** 7
**Confidence:** 4

**Strengths:**

Overall, I am quite positive on the paper.

I think the proposed captioning strategy is quite neat. The captions all manually seem quite sensible and strong (much stronger than just captioning a single image, using the metadata, and often human captions alone).

The approach is quite sensible, simple, and scalable as they can automatically be obtained on compute time alone without any human intervention. It nicely leverages other powerful models built by the community (i.e., BLIP, CLIP, GPT-4).

They provide nice evaluations and the paper is well written and easy to follow.

There is a need for strong text annotations for 3d, and this paper presents probably the best result I've seen in this space.

**Additional Feedback:**

I'm a bit confused why non-commercial assets are removed and unusable. Can you clarify this point? It seems like it would have been okay to use them for research purposes.

**Clarity:**

Yes, the paper is well written and easy to follow. Nice studies and comparisons. The appendix and datasheet are great.

**Correctness:**

The dataset is constructed in a sound way. I don't see any issues with respect to correctness.

**Documentation:**

The dataset is already released and linked, which is great. I do not have concerns with respect to a maintenance plan.

**Ethics:**

I don't have any ethical concerns that are novel to this paper. The ethics section is a nice touch.

**Limitations:**

It is not that fair of a baseline, but to me, this paper largely presents a workaround to caption objects with LLMs, given that LLMs largely don't have the ability to take in image input. It seems like once GPT-4 + vision is public, or other LLMs in this space that have the ability to consume images as input, the approach might not be that useful, and the captions that come out of GPT-4 + vision might be better than the captions that come out of Cap3D alone. Nevertheless, I don't think this should go against this work, as these models are not currently fully public.

**Opportunities For Improvement:**

W1. Captioning bias. A common criticism of using captioning models is that they often make the data much more biased than what comes from human captions, who often write in different styles, at different lengths, possibly in different languages, etc. Therefore,
W1.1. it would be nice to briefly compare captioning models in the full Cap3D pipeline against BLIP-2.
W1.2. I would also like to see something like a t-SNE plot of embeddings of human written captions compared to captions generated with Cap3D to see if the Cap3D ones are significantly biased towards a particular distribution.

W2. Captioning errors. It would be good to estimate the number of captions that are visibly incorrect, as they differ from the contents of the actual 3D object. Even a manual inspection estimate from <1K random objects would suffice here.

W3. I like the text-to-3D experiments idea, but the results are not that strong or significant. However, given this is a dataset submission, I won't hold it too much against the paper.

W4. Why choose 5 captions to generate? Do you consider other number of captions? Seems to me like only 2-3 captions would be sufficient in most cases.

**Relation To Prior Work:**

Yes, it points out where the objects come from, how it performs annotations, alternative pipelines and their cost.

**Summary And Contributions:**

The authors propose an approach for obtaining text captions of 3D objects.

It includes generating 5 captions using BLIP-2 from different rendered views of an object, choosing the best caption that most closely matches the view with CLIP, then using GPT-4, given the context of all the rendered view captions to create a single caption that best represents what's in the image.

They then perform several evaluation studies and find their captions are superior to other approaches, even including human written captions.

---

> ### Author Response · Authors · 2023-08-16
> **Author Response (1/3)**
>
> Thank you, Reviewer WJz6, for your insightful feedback and suggestions. We address your concerns below and sincerely look forward to your replies. We will incorporate our final discussion into the revision.
>
> ---
> Q1:
> > Captioning bias. A common criticism of using captioning models is that they often make the data much more biased than what comes from human captions, who often write in different styles, at different lengths, possibly in different languages, etc. Therefore, it would be nice to briefly compare captioning models in the full Cap3D pipeline against BLIP-2.
>
> A: Thanks for this suggestion. We first kindly refer to quantitative and qualitative comparisons between Cap3D and BLIP-2 in Table 3 and Figure 5 of the paper, including number of words per caption in Figure 5 (right). Notably, BLIP2 has little variance in the number of words per caption. Cap3D has a much higher variety in caption length, while increasing average caption length significantly. Also, during the rebuttal, we compared the number of n-grams. Cap3D has a much larger vocabulary than BLIP2.
>
> | Occ. in 5k captions | Unigrams | Bigrams | Trigrams |
> |---------------------|----------|---------|----------|
> | BLIP2               | 1,928    | 6,616   | 10,899   |
> | Cap3D               | 3,108    | 15,274  | 25,883   |
>
>
> ---
> Q2:
> > I would also like to see something like a t-SNE plot of embeddings of human written captions compared to captions generated with Cap3D to see if the Cap3D ones are significantly biased towards a particular distribution.
>
> A: Thank you for your suggestion. We conducted t-SNE over the sentence-embeddings of human-authored captions and Cap3D-generated captions for 37k objects. Results are [tsne-human](https://ltg-webaccess-files.s3.us-east-2.amazonaws.com/tsne_human.png) and [tsne-cap3d](https://ltg-webaccess-files.s3.us-east-2.amazonaws.com/tsne_Cap3D.png), which show Cap3D-generated captions are not significantly biased towards a particular distribution. Also, we put two sets of captions together before t-SNE. [tsne-fusion](https://ltg-webaccess-files.s3.us-east-2.amazonaws.com/tsne_fusion.png) shows the two distributions of captions are aligned to some levels.
>
>
> We further investigate by comparing caption length and dictionary size of Cap3D and human captions. In Figure 3 (right) of the paper, we show Cap3D caption length is actually typically longer than human caption length, while the distribution of caption lengths of Cap3D is flatter than human, meaning a bigger variety of caption lengths. Below we compare dictionary size. Although human captions use more phrases, the difference is much smaller than the difference between BLIP2 and Cap3D. We also hypothesize human crowdsourced captions contain more typos, leading to slightly higher dictionary size.
>
>
> | Occ. in 5k captions | Unigrams | Bigrams | Trigrams |
> |---------------------|----------|---------|----------|
> | Human               | 3,762    | 17,818  | 27,316   |
> | Cap3D               | 3,108    | 15,274  | 25,883   |
>
> ---
> Q3:
> > Captioning errors. It would be good to estimate the number of captions that are visibly incorrect, as they differ from the contents of the actual 3D object. Even a manual inspection estimate from <1K random objects would suffice here.
>
> A: Thanks for the suggestion, this can further evaluate the method’s ability. We manually evaluate 1000 random objects on a scale 1-3 (1: visibly incorrect, 2: minor mistakes, 3: fully correct), and find 29 (2.9%) are visibly incorrect, 118 (11.8%) have minor mistakes, and 853 (85.3%) are fully correct. Visibly incorrect examples tend to be on cluttered scenes or complicated architectures, such as that below.
>
> ***Uid***: fde49839440e4f4fa35fe497043d0a7f
>
> ***Image***: https://ltg-webaccess-files.s3.us-east-2.amazonaws.com/exp_r1q3.png
>
> ***Caption***: 3D rendering of a white shelf with a mirror, picture, and metal box, featuring a person standing near a window and a small snowy building, with a mountain and ceiling light

---

> > ### Author Response · Authors · 2023-08-16
> > **Author Response (2/3)**
> >
> > ---
> > Q4:
> > > I like the text-to-3D experiments idea, but the results are not that strong or significant. However, given this is a dataset submission, I won't hold it too much against the paper.
> >
> > A: Thank you for your recognition. We emphasize that the main contributions of our paper are a methodology for generating (text, 3D) datasets, along with a new public large-scale dataset of (text, 3D) pairs. Our primary aim in finetuning prior text-to-3D models on our dataset is not to achieve a new SOTA, but to (indirectly) demonstrate the utility and quality of our dataset compared to the private datasets used by OpenAI. A better demonstration would have been training text-to-3D models from scratch on Cap3D, but we unfortunately do not have computational resources for this; we chose finetuning as a more accessible proxy experiment.
> >
> > The current SOTA models for text-to-3D (Point-E and Shape-E) were trained on large, private datasets of (text, 3D) pairs of “several million 3D models and associated metadata” [1]; Shape-E augments this dataset with “roughly 1 million more 3D assets from high-quality data sources” and also “gathered 120K captions from human labelers for high-quality subsets of our dataset”.
> >
> > Finetuning Point-E on Cap3D significantly improves the model; this suggests that although the Point-E dataset is significantly larger than Cap3D, the improved text quality leads to improved model performance. Since the Shape-E dataset contains higher-quality human-written captions, we should not expect a large performance advantage when finetuning Shape-E on Cap3D; that we see roughly comparable performance suggests that the Shape-E dataset and Cap3D are perhaps of similar quality; however Cap3D was significantly less expensive to produce: per our estimates in Table 1, just the 120K human-written captions in the Shape-E dataset would cost $\sim$ 10,461 dollars, while the entire cost of all 660K samples in Cap3D is less than 5,000s dollar. These comparisons demonstrate the practical utility of Cap3D vs the private OpenAI datasets.
> >
> > [1] Point·E: A System for Generating 3D Point Clouds from Complex Prompts
> >
> > [2] Shap·E: Generating Conditional 3D Implicit Functions
> >
> >
> >
> >
> > ---
> > Q5:
> > > Why choose 5 captions to generate? Do you consider other number of captions? Seems to me like only 2-3 captions would be sufficient in most cases.
> >
> > A: Generating 5 BLIP-2 captions is a practical trade-off between computation cost and quality. Using 2-3 captions can result in mistakes and limited information, as we show below on our data using real outputs.
> >
> > ### Case1:
> > ***Uid***: b6b67e39466642dcb677629ae103de64
> >
> > ***Image***: https://ltg-webaccess-files.s3.us-east-2.amazonaws.com/exp_r1q51.png
> >
> > Below are the captions generated by BLIP for the (1st row, 3rd col) image. The first three captions contain inaccurate information, meaning if we use 2 or 3 of these captions we will obtain an inaccurate caption for this view. If we consider all 5 captions, we have 2 captions (4 and 5) which are more accurate, making it possible to select a good caption for this view, as CLIP does in this case (selected No. 4).
> >
> > ***Caption***:
> >
> >  [
> >  - '3d model of a bomb with a drink in it',
> >  - 'a black and yellow toy bomb with a pair of scissors',
> >  - 'a 3d model of a bomb with scissors',
> >  - 'a 3d model of a black and yellow bomb',
> >  - 'a 3d model of a bomb with a straw',
> > ]
> >
> >
> >
> > ### Case2:
> > ***Uid***: db83a97ab5ba4c7891f73ff9dea43714
> >
> > ***Image***: https://ltg-webaccess-files.s3.us-east-2.amazonaws.com/exp_r1q52.png
> >
> > Below are the captions generated by BLIP for the (2nd row, 1st col) image. Only the 2nd and 5nd one captions contain no false information, and CLIP selected 5nd one. Generating five captions per view offers CLIP diverse options to select from.
> >
> > ***Caption***:
> >
> > [
> > -  'an old book with a knife on top of it',
> > -  'a 3d model of a book with an eye on it',
> > -  'a 3d model of an old book on a gray background',
> > -  'an old book with a hole in it',
> > -  'a 3d model of an old book',
> > ]

---

> > > ### Author Response · Authors · 2023-08-16
> > > **Author Response (3/3)**
> > >
> > > ---
> > > Q6:
> > > > It is not that fair of a baseline, but to me, this paper largely presents a workaround to caption objects with LLMs, given that LLMs largely don't have the ability to take in image input. It seems like once GPT-4 + vision is public, or other LLMs in this space that have the ability to consume images as input, the approach might not be that useful, and the captions that come out of GPT-4 + vision might be better than the captions that come out of Cap3D alone. Nevertheless, I don't think this should go against this work, as these models are not currently fully public.
> > >
> > > A:  We agree GPT-4 + vision could be a good suggestion for future work and is not yet public. We note GPT-4 + vision is actually orthogonal to our contributions and could be used to improve our existing pipeline. Our contribution is to apply steps after image captioning in order to produce improved 3D captions. GPT-4 + vision could improve image captioning by replacing BLIP2, but it is likely GPT4 cross-image summarization could further improve outputs by aggregating information across views. GPT4 can also train on text-only data, which may supplement GPT-4+vision outputs.
> > >
> > > ---
> > > Q7:
> > > > I'm a bit confused why non-commercial assets are removed and unusable. Can you clarify this point? It seems like it would have been okay to use them for research purposes.
> > >
> > > A: Based on our experience, some industry groups would likely disallow the use of non-commercial assets even for research purposes out of an abundance of caution. As we are operating on a limited academic budget, we did not want to spend thousands of dollars running the Cap3D pipeline for commercial assets that would likely be filtered by a large cohort of potential downstream users of the dataset.

---

> > ### Comment · Reviewer_WJz6 · 2023-08-23
> > **brief response**
> >
> > Still going through the review, but just to clarify:
> >
> > What I meant by:
> >
> > > Therefore, it would be nice to briefly compare captioning models in the full Cap3D pipeline against BLIP-2.
> >
> > is that it would be nice to compare BLIP-2 to different captioning models in the Cap3D pipeline to see if better captioning models potentially help the bias issue.

---

> > > ### Author Response · Authors · 2023-08-25
> > > **Author response to "brief response"**
> > >
> > > Thank you for getting back to us. We look forward to your detailed response.
> > >
> > >
> > > ------
> > >
> > > For the below question
> > >
> > > > is that it would be nice to compare BLIP-2 to different captioning models in the Cap3D pipeline to see if better captioning models potentially help the bias issue.
> > >
> > >
> > > Thank you for the insightful question. We would like to address it by highlighting several key points:
> > >
> > > 1. **Consistency with Human-Generated Captions**: Based on the data from our study, the final captions that are a result of the integration of BLIP2, CLIP, and GPT4 do not exhibit a significant deviation from captions authored by humans. This observation is supported by Figure 3&4 in our paper, t-SNE results ([tsne-fusion](https://ltg-webaccess-files.s3.us-east-2.amazonaws.com/tsne_fusion.png), [tsne-human](https://ltg-webaccess-files.s3.us-east-2.amazonaws.com/tsne_human.png), [tsne-cap3d](https://ltg-webaccess-files.s3.us-east-2.amazonaws.com/tsne_Cap3D.png)), and our n-grams analysis.
> > >
> > >    Additionally, the complete pipeline we've implemented has shown promise in mitigating potential bias, as is evident from the n-grams analysis. Here, the Cap3D captions exhibit n-gram statistics more akin to human captions than to those of BLIP2 only:
> > >
> > > | Occ. in 5k captions | Unigrams | Bigrams | Trigrams |
> > > | ------------------- | -------- | ------- | -------- |
> > > | BLIP2               | 1,928    | 6,616   | 10,899   |
> > > | Cap3D               | 3,108    | 15,274  | 25,883   |
> > > | Human               | 3,762    | 17,818  | 27,316   |
> > >
> > > - **BLIP2's Superiority**: As of the current date (08/25/2023), BLIP2 stands as the state-of-the-art model in image-based captioning. We have yet to come across a model within the Cap3D suite that outperforms BLIP2, particularly with respect to addressing bias issues.
> > >
> > > - **The Potential of Future Models**: We recognize that advancements in image captioning will naturally benefit Cap3D. We are optimistic that superior captioning models will further enhance the quality of our Cap3D captions. As and when more robust image-based captioning models emerge, we are keen to incorporate them to deepen our understanding of the "bias issues". At this juncture, we'd appreciate your understanding in considering this as a prospective avenue for our future work.

---

> > > > ### Comment · Reviewer_WJz6 · 2023-08-25
> > > >
> > > > Thanks for clarifying and the pointers! That's very interesting. Do you think the added diversity in the captions comes mostly from GPT-4?
> > > >
> > > > The Table is very nice and surprises me.
> > > >
> > > > > https://ltg-webaccess-files.s3.us-east-2.amazonaws.com/tsne_fusion.png
> > > >
> > > > This plot is really neat! It'd be really interesting if you also plot the BLIP-2 captions. I suspect BLIP-2 will look just as diverse in this plot, but if it turns out that BLIP-2 looks more biased than the Cap3D and human captions, that'd be a really interesting and go well with the table!

---

> > > > > ### Author Response · Authors · 2023-08-25
> > > > > **Author Response**
> > > > >
> > > > > Thank you for your prompt feedback, Reviewer WJz6!
> > > > >
> > > > > We surmise that the observed increase in diversity stems from the consolidation of captions from eight different views provided by GPT-4.
> > > > >
> > > > > Thanks for the insightful suggestion. In response, we have produced a joint T-SNE visualization of human-author captions, Cap3D-generated captions, and BLIP2-generated captions. Please refer to [tsne-3-fusion](https://ltg-webaccess-files.s3.us-east-2.amazonaws.com/tsne_3_fusion.png). Upon observation, BLIP2 looks diverse generally; however, certain irregular distributions, notably in the top-right and bottom-left (which we've highlighted with green circles), are evident.

---

> > > > > > ### Comment · Reviewer_WJz6 · 2023-08-31
> > > > > >
> > > > > > Thanks for the results and quick responses. Overall, I believe the work should be accepted and will raise my score to a 7.
> > > > > >
> > > > > > I think the work provides a useful contribution to the community, both in terms of methodology of collecting the captions and in terms of actually having strong captions of 3D objects.

---

### Decision · Program_Chairs · 2023-09-22

**Decision:**

Accept (Poster)

**Comment:**

Five experts reviewed this paper with all accepted recommendations. The area chairs agree that this work makes a very important contribution by introducing a tool for 3D Captioning. The reviewers did raise some valuable concerns that should be addressed in the final camera-ready version of the paper. The authors are encouraged to make the necessary changes. It would be also nice if the authors could include some future work discussions on how to further make progress on this challenge by leveraging the recent success of concurrent works  (e.g. [1,2]) to further enhance the capability in the final version.

[1] 3D-LLM: Injecting the 3D World into a Large Language Model. arXiv:2307.12981

[2] PointLLM: Empowering Large Language Models to Understand Point Clouds. arXiv:2308.16911